

# Effects of wastewater treatment plant effluent inputs on planktonic metabolic rates and microbial community composition in the Baltic Sea.

Raquel Vaquer-Sunyer[1], Heather E. Reader[2], Saraladevi Muthusamy[3], Markus V.
Lindh[3§], Jarone Pinhassi[3], Daniel J. Conley[4] and Emma S. Kritzberg[5]
[1]{Interdisciplinary Ecology group, Department of Biology, University of the
Balearic Islands, 07122, Palma, Spain}
[2]{National Institute of Aquatic Resources, Section for Oceanography and Marine
Ecology, Technical University of Denmark, Charlottenlund, Denmark}
[3]{Centre for Ecology and Evolution in Microbial model Systems – EEMiS,
Linnaeus University, SE-39182 Kalmar, Sweden}
[4]{Department of Geology, Lund University, SE-223 62, Lund, Sweden}
[5]{Department of Biology, Lund University, SE-223 62, Lund, Sweden}
[§]{Present address: Department of Oceanography, Center for Microbial
Oceanography Research and Education (C-MORE), University of Hawaii at Manoa,
US-96822, Honolulu, USA}
Correspondence to: Raquel Vaquer-Sunyer, University of the Balearic Islands
(UIB). Crta. Valdemossa km 7.5, CP: 07122 Palma, Mallorca, Spain. Telephone:
(+34) 971172525, e-mail: raquel.vaquer@uib.cat

**Abstract**

The Baltic Sea is the world's largest area suffering from eutrophication-driven hypoxia. Low oxygen levels are threatening its biodiversity and ecosystem functioning. The main causes for eutrophication-driven hypoxia are high nutrient loadings and global warming. Wastewater treatment plants (WWTP) contribute to eutrophication as they are important sources of nitrogen to coastal areas. Here, we evaluated the effects of wastewater treatment plant effluent inputs on Baltic Sea planktonic communities in 4 experiments. We tested for effects of effluent inputs on chlorophyll a content, bacterial community composition, and metabolic rates: gross primary production (GPP), net community production (NCP), community respiration (CR) and bacterial production (BP). Nitrogen-rich dissolved organic matter (DOM) inputs from effluents increased bacterial production and decreased primary production and community respiration. Nutrient amendments and seasonally variable environmental conditions lead to lower alpha-diversity and shifts in bacterial community composition (e.g. increased abundance of a few cyanobacterial populations in the summer experiment), concomitant with changes in metabolic rates. An increase in BP and decrease in CR could be caused by high lability of the DOM that can support secondary bacterial production, without an increase in respiration. Increases in bacterial production and simultaneous decreases of primary production lead to more carbon being consumed in the microbial loop, and may shift the ecosystem towards heterotrophy.

**1   Introduction**

The Baltic Sea has the largest area affected by eutrophication-driven hypoxia (Conley et al., 2011). Eutrophication is expanding in the Baltic Sea; from 2007 to 2011 the entire open Baltic was found to be eutrophic (Fleming-Lehtinen et al., 2015). A 10-fold increase of the hypoxic area has been recorded for the last 115 years, mostly related to increased nutrient inputs from land (Carstensen et al., 2014). The lack of oxygen in marine waters causes death of marine organisms and catastrophic changes in marine metazoan communities. Thus, hypoxia is emerging as a major threat to marine biodiversity (Vaquer-Sunyer and Duarte, 2008), although prokaryotic diversity can increase in oxygen minimum zones (Wright et al., 2012).

Municipal wastewater treatment plants (WWTPs) contribute to eutrophication
because they are a substantial source of nitrogen (N) to natural waters worldwide
(Seitzinger *et al.* 2005). To reduce the environmental impact of WWTP effluent
discharge, limits on the concentration of nitrogen have been imposed. In the European
Union, 'the Urban Waste Water Directive' (91/271/EEC) sets the discharge limit of
effluents from urban wastewater treatment plants for total nitrogen (TN) between 10
and 15 mg N L$^{-1}$ (714 – 1071 µM), depending on the number of population
equivalents. In other regions, such as Chesapeake Bay, the largest U.S. estuary that
experiences severe hypoxic conditions, discharge limits range from 3 to 8 mg N L$^{-1}$
(214 – 571 µM, Chesapeake Bay Program 2006). Both areas, the Baltic Sea and
Chesapeake Bay, are enclosed water bodies with excessive anthropogenic nutrient
inputs. Wastewater treatment plants contribute 10-20% of total nutrient loading in the
Baltic Sea (Hautakangas et al., 2014). Estimates of total nitrogen loads to the Baltic
Sea due to WWTP effluents are about 110 000 tons of nitrogen per year, and for total
phosphorus loads are around 11 000 tones of phosphorus per year (Hautakangas et al.,
2014). Some Baltic countries have implemented nutrient reductions in their WWTP.
Denmark and Germany have reduced both nitrogen and phosphorus loadings
significantly. Sweden and Finland have reduced phosphorus loads but have failed so
far in reducing nitrogen loads down to 70% as recommended by HELCOM (2009)
(Hautakangas et al., 2014).
Effluent from WWTPs includes both dissolved inorganic (DIN) and organic N
(DON). The conventional biological treatment (secondary treatment) combines
coupled nitrification/denitrification and can potentially reduce TN to around 8-12 mg
N L$^{-1}$ (571-857 µM) (Bronk et al., 2010). Biological nutrient can eliminate most of the
DIN, leading to a substantial fraction of the residual N in effluent as DON (Bronk et
al., 2010; Grady et al., 2011). Effluents also contribute to increased organic matter
(OM) inputs to coastal areas.
DON can play an active role in providing nutrition to both phytoplankton and bacteria
(Berman and Bronk, 2003), and affects planktonic metabolism in areas receiving
significant amounts of DON. Dissolved organic matter (DOM) inputs to coastal areas
can also affect metabolic rates and favour bacterial processes (Berglund et al., 2007).
Here, we investigated the effects of wastewater treatment plant (WWTP) effluent
inputs on planktonic metabolic rates in the Baltic Sea. We did so on the basis of 4
experiments where WWTP inputs were added to natural communities. We tested for
effects of effluent inputs on metabolic rates: gross primary production (GPP), net
community production (NCP), community respiration (CR) and bacterial production
(BP); on chlorophyll a content; and on bacterial community composition.

## 2  Methods

### 2.1  Sampling

Natural marine planktonic communities from the Baltic Sea Proper were collected
(sampling dates included in Table 1) 10 km off the east coast of Öland, Sweden, at the
Linnaeus Microbial Observatory (LMO, N 56°55.851, E 17°03.640). The water was
sampled from 2 m depth and filtered through a 150 $\mu$m net to remove large grazers.
Wastewater effluent was collected within 10 days prior to experiment (sampling dates
included in Table 2) from the wastewater treatment plant (WWTP) in Kalmar for
effluent enrichment. Samples from WWTP were filtered using pre-combusted (450ºC,
4 h) glass-fiber (GF/F Whatman) filters and 0.2 $\mu$m membrane filters and frozen until
the start of the experiment. All equipment used for handling the samples was acid
washed.

### 2.2  Treatments

Four experiments were performed to cover all seasons: spring, summer, autumn and
winter, to be able to measure seasonal variation in both planktonic communities and
effluent characteristics under different environmental conditions. Each experiment
consisted of 5 different treatments: One with WWTP addition in a proportion of 1:10
vol:vol in seawater (1:10), a second with WWTP addition in a proportion of 1:5 (1:5);
a treatment with addition of inorganic nutrients (nitrate, nitrite and phosphate)
equivalent to that contained in the DON 1:5 treatment (IN). Those 3 treatments (1:10,
1:5 and IN) were performed to contain the same portion of community, so the 1:10
and the IN treatments were diluted with autoclaved milli-Q and salt solution to obtain
the same community portion than the 1:5 treatment. There was a control (C) treatment
with only seawater, and a diluted control (CD) consisting of seawater diluted with
autoclaved milli-Q water to have the same portion of community that the 1:10, 1:5
and IN treatments. To keep salinity constant in all treatments, a salt solution
(Søndergaard et al., 2003) was added with the amendments/dilutions.

## 2.3   Metabolic rates

Changes in dissolved oxygen (DO) in closed bottles were assumed to result from
biological metabolic processes and to represent net community production (NCP =
GPP – CR). Water from the respective treatments was siphoned carefully to avoid
bubble formation into four 2.3 L glass bottles per treatment sealed with gas tight
stoppers. Bottles were incubated at the in situ temperature (Tables 1 and S1) in a
temperature-controlled chamber during one week. Oxygen was measured every
minute in 2 of the 4 replicate bottles using optical oxygen sensors (optodes) and a 10-
channel fiber optic oxygen transmitter (oxy-10, PreSens®). The remaining 2 bottles
per treatment were used to sample for nutrient and chlorophyll a concentrations.
Incubations were illuminated by artificial light (OSRAM L36W/865 Lumilux
Daylight), with a PAR intensity of 1373.2 $\mu$W/cm$^2$. Light hours ranged from 8 h 30 m
on the winter experiment performed on January 2013 to 16 h 30 m on the summer
experiment on July 2013. This irradiation dose corresponds approximately to the
irradiation received at a depth of 2.5 m in the winter and 7 m in the summer, at
Kalmar, Sweden (Strång Model, SMHI).
NCP was estimated as the changes in DO content during 24 hours intervals (dDO/dt).
CR was calculated from the rate of change in DO during the night from half an hour
after lights went of to half an hour before light went on. CR was assumed to be the
same during light and dark. NCP in darkness equals CR during night. GPP was
estimated as the sum of NCP and CR (GPP = NCP + CR). Individual estimates of
GPP, NCP and CR resolved at one-minute intervals were accumulated over each 24-h
period during experiments and reported in mmol O$_2$ m$^{-3}$ day$^{-1}$, detailed description of
calculation of metabolic rates can be found at Vaquer-Sunyer et al. (2015).
As incubations were performed following a natural light regime to mimic natural
conditions, results may differ from incubations performed at light and dark conditions
in parallel. Both approaches assume equal respiration rates under light and dark
conditions. This assumption may lead to underestimate CR and GPP, as respiration
rates are probably higher during daylight than at night (Grande et al., 1989; Pace and
Prairie, 2005; Pringault et al., 2007), but it does not affect NCP estimates (Cole et al.,
2000). In incubations performed under dark conditions, phytoplankton growth is
suppressed, decreasing phytoplankton respiration contribution to community
respiration.

### 2.3.1 Bacterial Production

BP was estimated by measuring incorporation of $^3$H-leucine following the method
established by Smith and Azam (1992) on days 0, 1, 3, 5 and 7. Water samples (1.5
ml, 3 replicates and 1 killed control with 5% trichloroacetic acid (TCA)) were
incubated 60 minutes with 98.8 nM of $^3$H-leucine (13.4 Ci mmol$^{-1}$) in the
temperature-controlled room, at the same incubation temperature and light irradiance
as the rest of the samples. The incubation was terminated by adding TCA 5% final
concentration. The samples were then centrifuged at 16000g for 10 minutes and the
bacterial pellet was washed once with 5% TCA and once with 80% ethanol. After the
supernatant was discarded, 0.5 ml of scintillation cocktail (Ecoscint A, Kimberly
Research) was added and $^3$H -activity measured on a Beckman LS 6500 scintillation
counter. BP was calculated according to Smith and Azam (1992) assuming a leucine
to carbon conversion factor of 1.5 kg C mol$^{-1}$ leucine (Kirchman, 2001).

### 2.4 Chlorophyll a, dissolved organic carbon and nutrient measurements

Samples for chlorophyll a (*Chl.a*), dissolved organic carbon (DOC) and nutrients
were taken on days 0, 1, 3, 5 and 7 from the two 2.3 L bottles for each treatment
incubated in parallel with the bottles used to monitor oxygen changes. Samples were
taken in duplicate. For the last day of the experiment (day 7) the 2 bottles used to
monitor oxygen content were used to sample *Chl.a*, DOC and nutrient content.
Samples for nutrient determination were filtered using pre-combusted (450ºC, 4 h)
glass-fiber (GF/F Whatman) filters and 0.2 $\mu$m membrane filters and frozen until
analysis. All equipment used for handling the samples was acid washed.
Chlorophyll a was measured in duplicate following Jespersen and Christoffersen
(1987) on a Turner TD-700 fluorometer.
DOC was measured on a Shimadzu TOC V-CPN in non-purgeable organic carbon
(NPOC) mode on acidified samples (HCl to pH < 2). The instrument was calibrated
daily with potassium hydrogen phthalate. DOC concentrations were calculated from
the average area of 3 injections, with an area covariance of less than 2%.
Total dissolved nitrogen (TDN) was measured in duplicate after persulfate oxidation.
The method of persulfate oxidation was chosen instead of high temperature combustion
(HTC), as it has been demonstrated to be more appropriate for eutrophic waters, such as
the Baltic Sea, as well as coastal areas (Bronk et al., 2000). Inorganic nutrient analyses
(nitrate ($NO_3^-$), nitrite ($NO_2^-$) and phosphate ($PO_4^{3-}$)) were analysed in duplicate on an
automated nutrient analyser SmartChem® 200. Concentration of ammonium ($NH_4^+$)
was measured in duplicate on a spectrophotometer following the manual phenol
hypochlorite method by (Koroleff, 1983). The concentration of DON was calculated by
difference after subtracting the concentration of $NH_4^+$, $NO_3^-$, and $NO_2^-$ from the TDN
concentration. Dissolved primary amines (DPA) concentrations were measured in
triplicate on a spectrofluorometer following the OPA (*o*-phthaldialdehyde) method
(Parsons et al., 1984).

## 199 2.5 Bacterial Diversity

Bacterial 16S rRNA gene fragments were amplified with bacterial primers 341F and
805R (Herlemann et al., 2011) following the PCR protocol of Hugerth et al. (2014)
with some modifications. We thus performed a two-step PCR: (i) amplification with
the main forward and reverse primers 341F-805R to amplify the correct fragment
within the V3-V4 hypervariable region of the 16S rRNA gene; (ii) amplification using
template from the first PCR to attach the handles and indexes needed to run the
Illumina Miseq run and for barcoding individual samples. Amplification was carried
out in duplicates for each biological replicate using an annealing temperature of 58°C
in the first PCR and 12 cycles in the second PCR. The resulting purified amplicons
were sequenced on the Illumina Miseq (Illumina, USA) platform using the 300 bp
paired-end setting at the Science for Life Laboratory, Sweden (www.scilifelab.se).
Raw sequence data generated from Illumina Miseq were processed using the
UPARSE pipeline (Edgar, 2013). Taxonomy was determined against the
SINA/SILVA database (SILVA 115; Quast et al., 2013). After quality control, our
data consisted of a total of 3.8 million reads, with an average of 68218.61 ± 33048.86
reads per sample. These sequences resulted in a final OTU table consisting of 3420
OTUs (excluding singletons) delineated at 97% 16S rRNA gene identity. DNA
sequences have been deposited in the National Center for Biotechnology Information
(NCBI) Sequence Read Archive under accession number SRP059501.

## 2.6 Statistics

Relationships between chlorophyll a contencentration and physicochemical parameters (nitrate concentration, light hours and temperature) were tested by fitting ordinary least square regression.

Metabolic rates data from the four experiments were combined to test the relationship between the given metabolic rates and physicochemical parameters (Table 1) by mixed effects models. Physicochemical parameters were chosen avoiding collinearity. Selected variables were DOC, DON, nitrate and phosphate concentration. We used DOC as a proxy for dissolved organic matter (DOM). Variables were selected according to its significance. Variables were removed from the model following its p value (i.e. variables with higher p value were removed first) until all variables were significant. To account for pseudo-replication we used incubation day nested to season (i.e. experiment) as a random factor. The pseudo-$R^2$ of the models was calculated following Xu (2003).

Differences in community composition between treatments were tested using permutational analysis of variance (PERMANOVA) on Bray-Curtis distances. To test the correlation between absolute changes in environmental conditions, metabolic rates and absolute shifts in bacterioplankton community composition we performed MANTEL tests. For alpha-diversity measures we subsampled each sample to 10 000 sequences. Analyses performed at the OTU level were based on selecting the top 200 most abundant OTUs. For OTU level analyses on Cyanobacteria we selected OTUs affiliated with Cyanobacteria among the top 200 most abundant OTUs. Taxonomic annotation from SINA/SILVA database was limited for cyanobacterial OTUs and we therefore extended the annotation by using BLASTn (NCBI). For all analyses on community composition we examined the following major eight phyla/classes: Actinobacteria, Bacteroidetes, Alphaproteobacteria, Betaproteobacteria, Gammaproteobacteria, Cyanobacteria, Planctomycetes, and Verrucomicrobia. All other phyla/classes were grouped together and defined as "Others". All statistical tests were performed in R 3.0.2 (R Core Team, 2014) and using the package Vegan (Oksanen et al., 2010). Graphical outputs were made using the package ggplot2 (Wickham, 2009). Phylogenetic analyses using maximum likelihood trees were performed with MEGA 6.0.6 and the Tamura-Nei model (Tamura et al., 2011).

## 3    Results

Treated wastewater nutrient content differed between seasons (Table 2). The highest TDN values were measured in winter (600.1 ± 6.6 µM), whereas the lowest values were measured in summer (518.4 ± 2.4 µM). DON content in wastewater effluent varied between 75.2 ± 4.4 µM in autumn and 503.3 ± 2.9 µM during winter. The DOC:DON ratio was low (2.1 – 9.4), indicating nitrogen rich dissolved organic matter (DOM). In summer and spring phosphate content in the effluent was below detection limit (30 µg/L, Table 2).

Nutrient content in the seawater also differed between seasons (Table 1), with the highest TDN value in autumn (21.0 ± 0.30 µM), and the lowest values were measured in spring (16.4 ± 0.6 µM). DON content in coastal water ranged between 11.4 ± 0.9 µM and 17.9 ± 0.5 µM, measured in winter and autumn respectively.

### 3.1    Chlorophyll a

Coastal waters showed a typical seasonal pattern (Vahtera et al., 2007), with low chlorophyll a (*Chl.a*), and high nutrient content in winter; in spring, with the increase in solar radiation, *Chl.a* increased, and inorganic nutrients started to decrease. In summer with high temperature and high sunlight radiation, *Chl.a* values increased to the maximum measured, and inorganic nutrients were depleted (Table 1). During autumn, *Chl.a* content decreased to the second lowest values and nutrient concentration started to replenish (Table 1).

Chlorophyll a content strongly depended on light availability ($p < 0.0001$, $R^2 = 0.60$) and on temperature ($p < 0.0001$, $R^2 = 0.41$), with the summer experiment having the highest values (Mean ± SE = 7.59 ± 0.41 µg $L^{-1}$), with 16.5 light hours and a mean temperature of 18.4 ℃ (Fig.1, Supplementary Information (SI) Table S1). 66% of *Chl.a* variation could be explained by changes in light exposure time and $NO_3^-$ concentration ($p < 0.0001$).

### 3.2    Metabolic Rates

### 3.2.1    Gross Primary Production

Gross primary production (GPP) for natural communities in the experiments varied from 2.03 ± 2.00 to 54.16 ± 5.31 mmol $O_2$ $m^{-3}$ $d^{-1}$, both extremes measured on the 5th

day of the experiment, for experiments conducted in winter and summer, respectively.
In the amended treatments, GPP also varied greatly between days of experiment and
seasons, with the lowest measured GPP being $0.14 \pm 1.91$ mmol $O_2$ m$^{-3}$ d$^{-1}$ for the 5$^{th}$
day of the 1:10 treatment in the experiment conducted in winter; and the highest
measured GPP was $85.67 \pm 7.13$ mmol $O_2$ m$^{-3}$ d$^{-1}$ on the final day (day 7) of the
inorganic nutrient addition treatment in summer (fig. 2).
GPP variability was explained by differences in DOC concentration (Table 3), with
this variable explaining 84% of its variability (fig 3a). GPP decreased with DOC
concentration (Table 3).
**3.2.2  Community Respiration**
Community respiration (CR) for natural waters in the experiments varied between
$5.30 \pm 0.99$ and $34.89 \pm 1.35$ mmol $O_2$ m$^{-3}$ d$^{-1}$ (Table S1). CR varied greatly between
treatments, days of experiment and seasons. CR varied from $0.95 \pm 1.32$ mmol $O_2$ m$^{-3}$
d$^{-1}$ for the day 1 on the IN treatment from the winter experiment to $54.16 \pm 55.59$
mmol $O_2$ m$^{-3}$ d$^{-1}$ for the final day on the 1:5 treatment during the fall experiment (fig.
4). The high SD associated to these measures is due to differences between incubation
bottles.
CR was inversely correlated to DOC concentration, with this variable explaining the
84% of CR variability (Table 3, fig. 3b).
**3.2.3  Net Community Production**
Net community production (NCP) for natural communities in the experiments varied
between -8.83 and $20.17 \pm 5.78$ mmol $O_2$ m$^{-3}$ d$^{-1}$ measured on fall and on summer,
respectively. The range of variability in the treatments with nutrient additions was
wider ranging from $-16.64 \pm 17.69$ to $36.69 \pm 1.49$ mmol $O_2$ m$^{-3}$ d$^{-1}$ measured in the
day 1 on the 1:10 treatment in the winter experiment and in day 7 on the IN treatment
during the summer experiment, respectively (fig. 5). NCP varied greatly between day
of experiment, season and treatment.
NCP was dependent on DOC concentration, with this variable explaining the 79 % of
its variability (Table 3, fig. 3c). NCP significantly decreased with DOC content ($p <$
$0.0001$, Table 3).

### 3.2.4  Bacterial Production

Bacterial production (BP) tended to increase in the treatment with the higher addition of effluent (fig. 6). Repeated measures MANOVA showed significant differences in BP for different sampling days, for treatments and for the interaction between sampling day and treatment for experiments conducted in summer and fall ($p < 0.0001$ for both cases). Conversely, BP was not significantly different between treatments for experiments conducted in spring and winter. For those experiments there were significant differences in BP between sampling days and in the interaction between treatment and sampling day.

BP was positively correlated to DOC content in spring, summer and winter ($p < 0.003$, $p < 0.005$ and $p < 0.05$, respectively), but it was independent of DOC concentration in fall ($p > 0.05$).

The variables that best explained BP variability were phosphate, DOC, DON and $NO_3^-$ concentration ($R^2 = 0.91$, Table 3, fig. 3d). BP increased with DOC, DON and nitrate concentration and decreased with phosphate concentration.

### 3.3  Bacterial diversity and community composition

Bacterial community structure showed two distinct clusters with summer communities separated from spring and winter across all experiments (fig. S1, Supplementary Information). Community composition in each experiment exhibited, in general, a temporal succession and an additional response to different treatments. We carried out MANTEL tests to elucidate the influence of environmental factors on community composition and metabolic rates. Changes in temperature significantly explained absolute shifts in bacterioplankton community composition across all experiments (Pearson $r > 0.5$; Table 4). Changes in GPP, CR, BP, Chl *a*, $NO_2^-$ and $PO_4^{3-}$ were significantly correlated with absolute shifts in bacterioplankton community composition, with the highest correlation observed for $PO_4^{3-}$ (Pearson $r = 0.30$; Table 4).

Alpha diversity estimated from Shannon index was relatively similar between treatments in each experiment and ranged from $3.34 - 5.82 \pm 0.51$ (fig. 7). Nevertheless, a lower Shannon index was observed for all nutrient treatments compared to the controls in all experiments except April (fig. 7). Moreover, we analysed the richness and found that the observed number of OTUs ranged between

206-946 ± 171 and Chao.1 index values ranged between 306-1273±220 (fig. S2).
Richness was generally lower in effluent amended treatments compared to controls,
except for in the April experiment.
Betaproteobacteria, Bacteroidetes and Alphaproteobacteria dominated the April
experiment where Betaproteobacteria displayed a marked increase in relative
abundance from T0 to T7 (Fig. 8). In general, few differences in community
composition between treatments were observed. Nevertheless, Betaproteobacteria
decreased in relative abundance by more than half in controls until T7 while they
maintained their abundance in the other treatments. For the January experiment
differences between treatments were more pronounced (fig. 8). Bacterial groups other
than the 8 major phyla/class ("Others") had nearly four-fold higher relative abundance
in the 1:5 treatment compared to the other treatments and the controls. At T3
Cyanobacteria had considerably higher relative abundance in the 1:10 and IN
treatments compared to the controls and 1:5 treatment. The July experiment showed a
higher relative abundance of Cyanobacteria and Verrucomicrobia, with the relative
abundance of Cyanobacteria increasing over time in the amended treatments. In
contrast, the relative abundance of Verrucomicrobia increased in the control
treatments and was highest in the diluted control (CD) (fig. 8). Hence, Cyanobacteria
had higher relative abundance in treatments with additions of nutrients (both DON
and IN; fig. 8). For the November experiment there was an overall greater variation in
community composition. Still, relative abundances of Gammaproteobacteria increased
in the IN treatments at T3 and T7 compared to the other treatments and control.

## 366    3.4   Population dynamics

Patterns in community composition indicated that effluent amendments had an effect
on bacterial population dynamics in our experiments coupled with the concomitant
changes in metabolic rates. Hence, we performed Pearson correlation tests to
determine links between environmental factors, metabolic rates and shifts in relative
abundances at phyla/class level. Shifts in relative abundances of Cyanobacteria,
Planctomycetes and Verrucomicrobia were positively correlated with temperature
(fig. 9). In contrast, Alphaproteobacteria, Bacteroidetes and Betaproteobacteria were
negatively correlated with temperature. Cyanobacteria, Planctomycetes and
Verrucomicrobia displayed a strong negative correlation with community respiration
but a positive correlation with bacterial production. These three groups of bacteria
were also negatively correlated with $PO_4^{3-}$ while Alphaproteobacteria, Bacteroidetes
and Betaproteobacteria were positively correlated with $PO_4^{3-}$. In particular, changes in
$PO_4^{3-}$ concentrations explained > 50 % of the variance for Bacteroidetes (fig. 9). In
addition, Verrucomicrobia had a strong correlation with $NO_2^-$. Actinobacteria,
Gammaproteobacteria and bacterial groups other than the 8 major phyla/class
("Others") showed only weak correlations with environmental parameters and
metabolic rates.
Changes in relative abundance of particular bacterial populations typically followed
the overall pattern within each major phyla/class. For example *Chtoniobacterales*
OTUs within Verucomicrobia exhibited positive correlations with temperature and
bacterial production but negative correlations with $PO_4^{3-}$ (fig. S3). Although relative
abundances of Gammaproteobacteria showed overall weak correlations with
metabolic rates and environmental factors, the relative abundance of specific OTUs in
this taxon, such as OTU 001410 and two *Halioglobus* OTUs (OTU 001149 and OTU
000045), displayed strong correlations (Pearson's r >0.5) with temperature, bacterial
production and community respiration. Betaproteobacteria OTUs showed overall
weak correlations with metabolic rates and environmental factors except for two
MWH-UniP1 related OTUs (OTU 002372 and OTU 000041). Betaproteobacteria
affiliated with BAL58 showed in some cases a substantial correlation (Pearson's r >
0.5) with DOC (OTU 001633, OTU 001481, OTU 000008 and OTU 001907) (fig.
S3). Within Alphaproteobacteria most OTUs had weak correlations. However, one
particular alphaproteobacterial OTU affiliated with Rhodobacteraceae (OTU 000044)
exhibited strong correlations with metabolic activities and environmental variables,
both negative (e.g. $PO_4^{3-}$ and community respiration) and positive (e.g. temperature
and bacterial production). Moreover, 10 *Rhodobacteraceae* OTUs were positively
correlated with DOC. *Synechococcus* OTUs were positively correlated with
temperature, NCP, GPP, bacterial production and Chl *a* (fig. S3).
To extend the analysis of the strong Cyanobacteria population dynamics observed in
the July experiment, we investigated particular OTUs and plotted relative abundances
of this group across all experiments (fig. S4). For the other experiments,
cyanobacterial populations had, in general, low relative abundance but were still more
abundant in treatments with effluent and nutrients amendments than without (except
for the April experiment). Six OTUs showed particularly high relative abundance in
the July experiment (fig. S4). These cyanobacterial populations increased with time
and at T7 both *Synechococcus* and *Cyanobium* populations had higher relative
abundance in treatments of 1:10, 1:5 and IN compared to controls.

## 4 Discussion

Nitrogen-rich dissolved organic matter (DOM) from WWTP effluents had significant
impacts on Baltic Sea planktonic metabolic rates: DOM significantly increased
bacterial production, whereas it decreased gross and net primary production and
community respiration rates, as showed in the results of the mixed effects models
where DOC is used as a proxy for DOM. Bacterial production was also positively
correlated to DON concentration, supporting that DON can provide nitrogen nutrition
to bacteria. BP was negatively correlated to phosphate concentration, due to seasonal
variations, as phosphate content is higher in winter when BP is low. A parallel
increase in BP and decrease in bacterial respiration (BR) rates results in an increase in
bacterial growth efficiency (BGE = (BP)/(BP + BR), (del Giorgio and Cole, 1998)).
Literature values for BGE in the Baltic Sea vary substantially from 0.06 to 0.6 (Donali
et al., 1999). Here we did not measure bacterial respiration separately, but as a part of
total community respiration. Assuming that bacterial respiration contributes 50% of
community respiration (Williams, 1981; Aranguren-Gassis et al., 2012) we can
estimate BGE. As BR is known to be higher than 50% of CR (Williams, 1981), this
approach will result in an underestimation of bacterial growth efficiency but will
suffice to support our hypothesis that DOM additions increased BGE. Estimated BGE
for our experiments varied between 0.06 and 0.59, consistent with previous reported
values (Donali et al., 1999; Zweifel et al., 1993). Estimated BGE increased with
nitrate ($p < 0.003$) and DOC concentration ($p < 0.0007$) and decreased with phosphate
content ($p < 0.02$, mixed effects model, $R^2 = 0.78$). An increase of BGE with nutrient
addition was reported for communities from the Bothnian Bay, increasing from a
range of 0.11 - 0.54 to 0.14 - 0.58 for treatments with nutrient amendment (Zweifel et
al., 1993). Other studies also report an increase in BGE with DOM and nutrient
additions in three estuaries from the Baltic Sea (Asmala et al., 2013). Our estimation
of BGE shows a positive effect of N-rich DOM on bacterial growth efficiency,
suggesting high lability of N-rich WWTP effluent DOM, where most of the carbon
can be used for secondary bacterial production and a low portion is respired.
Wastewater treatment plant effluent inputs to the Baltic Sea raised bacterial
production at the same time as it reduced primary production, leading to more carbon
being used by the microbial loop. This increase in bacterial production parallel with a
decrease in primary production moves the ecosystem towards heterotrophy. This is
supported by a higher BP:NCP ratio in treatments with addition of WWTP effluent
(mean = 1.56 ± 0.38), compared to treatments without amendment (mean = 0.66 ±
0.32), although this differences are not significant ($p > 0.05$). Increased flow of
organic matter through the microbial loop could result in a reduction of the transfer of
carbon to higher trophic levels and of the efficiency of the biological carbon pump in
sequestering carbon (Berglund et al., 2007; Wohlers et al., 2009). Bacteria-based food
webs generally have lower food web efficiency due to the smaller sizes of the
resources and predators, leading to more trophic levels than phytoplankton-based food
webs. As around 70% of ingested carbon is lost at each trophic level due to respiration
and sloppy feeding (Straile 1997), larger carbon losses are expected in bacteria-based
food webs (Berglund et al., 2007). Whereas some studies suggest that an increased
flow of carbon through the microbial loop would result in a reduction of the biological
carbon pump efficiency in sequestering carbon, a recent study suggests the opposite:
marine bacteria can produce refractory exometabolites that would result in carbon
sequestration (Lechtenfeld et al., 2015).
Effluent inputs decreased GPP and NCP, resulting in a reduction of photosynthetic
rates, declining oxygen production in the photic layer. The Baltic Sea is already the
largest eutrophication-driven hypoxic area in the world (Conley et al., 2011), and a
decrease of biological oxygen production could further aggravate hypoxic conditions
in this already affected area. The lack of oxygen is an important environmental
problem is this area, it produces a reduction of marine benthic diversity as a result of
the death of sensitive marine organisms and it affects biogeochemical cycles (Conley
et al., 2009). It increases phosphorus fluxes from sediments into overlaying waters,
changing redox conditions in the water column and reduces the ecosystem capacity of
removing nitrogen, as a consequence of the reduction of the substrate needed for
denitrification (nitrate) when sediments become more reducing (Conley et al., 2009).
Although several microbial taxa showed weak correlations with contemporary
changes in environmental conditions and/or metabolic activity, specific opportunistic
populations proliferated in effluent input treatments. In particular, verrucomicrobial
and cyanobacterial populations responded in relative abundance to effluent inputs in
summer. Thus, OTUs affiliated with Verrucomicrobia decreased in relative
abundance in the treatments with effluent addition compared to controls. In contrast,
the relative abundance of a few specific cyanobacterial populations increased upon
enrichment (but less so in controls, i.e. the cyanobacterial growth was not only an
effect of higher temperatures in the summer experiment). Generally, it is likely that
the proliferation of cyanobacteria in the summer experiment is linked to the actual
abundance of cyanobacteria, which is typically higher in summer, so that the
"seeding" population for this taxon was higher. The Baltic Sea suffers from extensive
Cyanobacteria blooms in summer that can easily be observed from space, primarily
caused by eutrophication (Vahtera et al., 2007). The death and sedimentation of
Cyanobacteria blooms, and the subsequent decay of this organic material is a
contributing mechanism for oxygen depletion in bottom waters. Consequently,
Cyanobacteria blooms have been linked to hypoxia development and expansion in the
Baltic Sea. Warming could further increase cyanobacteria blooms in the Baltic Sea
(Paerl and Huisman, 2008; Paerl and Paul, 2012). Here, we found that relative
abundances of Cyanobacteria were positively correlated with temperature.
Links between metabolic activity and compositional changes of bacterial communities
are frequently observed in aquatic ecosystems (Bell et al., 2005; Allison and Martiny,
2008; Logue et al., 2016). Yet, in other cases, such linkages are relatively weak and
possibly confounded by environmental complexity (Comte and Del Giorgio, 2011;
Comte et al., 2013; Langenheder et al., 2005; Langenheder et al., 2010). Our results
showed that effluent inputs caused simultaneous shifts in community composition
coupled with changes in metabolic rates. Changes in temperature were the major
driver of community structure but also phosphate significantly explained variations in
the relative abundance of particular groups and taxa. This emphasizes that changes in
temperature and nutrient availability can affect bacterioplankton community
dynamics. Similarly, differences in temperature and nutrient conditions lead to shifts
in community structure in for example mesocosm experiments with Mediterranean
and Baltic Sea microbial assemblages (Degerman et al., 2013; Gomez-Consarnau et
al., 2012; Pinhassi et al., 2006; von Scheibner et al., 2014). More importantly, in these
studies, compositional shifts occurred with concomitant responses in community
metabolic activity. Apart from the influence of temperature in structuring the bacterial
communities in the present study, shifts in bacterioplankton community composition
were highly correlated with changes in phosphate concentrations. In agreement,
previous findings show that phosphate is a driver of shifts in community structure in
the Southern Californian coast and Baltic Sea (Fuhrman et al. 2006; Andersson et al.
2010). For example, Andersson and colleagues (2010) suggested that limiting
conditions due to a decline in phosphate during the summer Cyanobacterial bloom
promote selection in the bacterioplankton community where specific OTUs can
proliferate. Moreover, in an adjacent area of the Baltic Sea Proper opportunistic
cyanobacteria, including $N_2$-fixers and picocyanobacteria, proliferated despite low
phosphorus concentrations and may instead have been fueled by bioavailable nutrients
from filamentous Cyanobacteria (Bertos-Fortis 2016). Recent evidence suggests that
availability of phosphorus has a substantial impact on eutrophication in the Baltic Sea
since many Cyanobacteria are able to fix nitrogen (Andersson et al. 2015). In the
present study phosphate concentrations showed small variations between treatments
within each experiment and we observed primarily seasonal oscillations between
experiments. Absolute shifts in composition among the groups Bacteroidetes,
Betaproteobacteria and Alphaproteobacteria were positively correlated with absolute
changes in phosphate whereas shifts in Planctomycetes, Verrucomicrobia and
Cyanobacteria were negatively correlated with variation in phosphate. Nevertheless,
changes in phosphate concentrations significantly explained variation in community
structure within the July experiment. Hence, the communities responded to effluent
inputs by shifts in species composition and the influence of seasonal changes in
phosphorus concentrations was outweighed by the simulated environmental
disturbance investigated here. Thus, long-term changes in phosphorus resulting from
natural seasonal variation or climate change related effects accompanied by episodic
short-term effluent inputs may form a synergistic permanent impact on the structure
of bacterioplankton communities with severe consequences for ecosystem services. In
agreement, shifts in community composition can be closely linked with changes in
community functioning, i.e. metabolic rates, (e.g. Bell et al. 2005; Allison and
Martiny 2008). In addition, alpha-diversity was lower in effluent input treatments.
The observed effect of species loss, i.e. lower richness (observed number of OTUs
and Chao.1 index) and Shannon diversity index, may be closely linked with the
functioning of microbial communities and could potentially render the whole
community more sensitive to environmental perturbations (Allison and Martiny,
2008; Bell et al., 2005; Loreau, 2000, 2004; Shade et al., 2012). Alternatively, lower
richness and Shannon diversity index does not necessarily implicate loss of
community functioning as previously observed in e.g. lake systems (Comte and del
Giorgio 2011; Langenheder et al. 2005). Hence, our findings suggest that linked
alterations in bacterial community composition and metabolic activity from
anthropogenic changes could potentially affect biogeochemical cycling of elements in
the coastal Baltic Sea.
The so-called "bottle-effect", in which confinement of water causes shifts in
bacterioplankton community composition and physiological rates, is a factor to
consider in interpreting results from experiments with natural microbial assemblages
(Fuchs et al., 2000; Massana et al., 2001; Baltar et al., 2012). Such effects are
typically detected by rapidly increasing proportions of fast-growing
gammaproteobacterial populations and rate measurements across all treatments
(including controls) (Pinhassi and Berman, 2003; Sjöstedt et al., 2012; Dinasquet et
al., 2013). In our current experiments, microbial community composition remained
relatively similar to in situ communities and we did not observe excessive increases in
opportunistic bacterial populations in the controls. Rather, increases and decreases in
relative abundance were observed among populations typical of Baltic Sea Proper,
such as *Rhodobacteraceae*, *Synechococcus* and BAL58 (Lindh et al., 2015). Thus,
although confinement per se surely had effects on microbial diversity and rates, our
results indicate that such effects were minor relative to the actual treatment effects.
Inputs of WWTP effluent in summer further stimulated bacterial production, when it
was already high due to elevated temperatures. Summer was the period of the year
that responded sharply to effluent additions. Warming could also increase respiration
rates to a larger degree than primary production, moving the system towards
heterotrophy (Brown et al., 2004; Harris et al., 2006; Vaquer-Sunyer et al., 2015;
Yvon-Durocher et al., 2010). Simultaneous warming and inputs from wastewater
treatment plant effluents increased planktonic respiration rates and bacterial
production faster than it increased planktonic primary production in the Baltic Sea
(Vaquer-Sunyer et al., 2015), leading to higher biological oxygen consumption than
production, which may lead to the depletion of the oxygen pool, further aggravating
hypoxia in the Baltic Sea. Here, we found that WWTP effluent inputs increased
bacterial production at the same time that decreased net and gross primary production

and community respiration. A parallel increase in bacterial production and decrease in primary production leads to more carbon being used by the microbial loop and may have consequences on the food web transfer efficiency.

## 5    Conclusions

The current study showed that inputs of DOM from WWTP effluents were related to increased bacterial production and decreased primary production and community respiration, which could lead to an increase in BGE. DON concentration enhanced bacterial production, suggesting that bacteria can use DON as nitrogen source. The increase in BP and decrease in CR could be caused by high lability of the OM that supported secondary bacterial production, without an increase in respiration. Seasonal changes in temperature were the most important factor for structuring community composition but also phosphate concentrations significantly explained variations in the relative abundance of particular groups and taxa. In summer, the relative abundance of Cyanobacteria increased after effluent inputs (but less so in the controls). Cyanobacteria have been linked to hypoxia in the Baltic Sea, and an increase in their abundance could result in oxygen depletion of the Baltic bottom waters. Inputs from wastewater treatment plant effluent could further worsen hypoxic conditions in the Baltic Sea.

Reductions of the OM content in wastewater treatment plant effluents are needed to reduce its potential negative consequences. Effluent inputs resulted in a reduction of photosynthetic rates, moving the system towards heterotrophy, decreasing oxygen production in the photic layer in the Baltic Sea.

**Authors contributions**

RVS designed research and performed experiments. ML, JP and SDM analysed bacterial diversity samples and data. HER wrote the code for metabolic rates calculations. All authors were involved in the writing stage of the manuscript and collaborated on the analysis, interpretation, and discussion of the results.

**Acknowledgements**

The authors would like to thank Catherine Legrand, Emil Fridolfsson, Anders Månsson and Kristofer Bergström at the Linnaeus University for their help on

Kalmar's WWTP effluent water and seawater sampling. We would also like to thank
Carolina Funkey for help with nutrients analysis and E. Baraza for statistical advice.
RVS was supported by a Marie Curie Intra European Fellowship (IEF). This research
is a contribution to the "The role of dissolved organic nitrogen (DON) on the
development and extent of eutrophication-driven hypoxia and responses to global
warming" project, funded by the FP7 Marie Curie IEF, grant number 299382 and to
the "Managing multiple stressors in the Baltic Sea" project, funded by FORMAS,
grant number 217-2010-126. This work resulted from the BONUS COCOA project
and the BONUS BLUEPRINT project that were supported by BONUS (Art 185),
funded jointly by the EU and Swedish Research Council FORMAS.

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

**Tables**
Table 1. Physicochemical parameters in coastal seawater for the different sampled
seasons. Standard errors (SE) are derived from duplicate sample analysis. C:N ratio is
calculated as the ratio DOC:DON (moles).

| | Winter | Spring | Summer | Autumn |
|---|---|---|---|---|
| Date | 23/01/2013 | 03/04/2013 | 18/07/2013 | 04/11/2013 |
| TDN (± SE) (µM) | 17.01 (±0.87) | 16.40 (±0.63) | 16.51 (±0.08) | 20.99 (±0.34) |
| $NO_2^-$ (± SE) (µM) | 0.35 (±0.02) | 0.14 (±0.00) | 0.09 (±0.01) | 0.31 (±0.21) |
| $NO_3^-$ (± SE) (µM) | 4.93 (±0.39) | 3.69 (±0.14) | 0.50 (±0.09) | 2.64 (±0.32) |
| $NH_4^+$ (± SE) (µM) | 0.35 (±0.01) | 0.01 (±0.01) | 0.24 (±0.00) | 0.23 (±0.03) |
| $PO_4^{3-}$ (± SE) (µM) | 0.55 (±0.03) | 0.63 (±0.03) | 0.03 (±0.01) | 0.39 (±0.02) |
| DON (± SE) (µM) | 11.44 (±0.95) | 12.56 (±0.64) | 15.76 (±0.12) | 17.91 (±0.47) |
| DPA (± SE) (µM) | 0.09 (±0.01) | 0.31 (±0.01) | 0.17 (±0.01) | 0.24 (± 0.03) |
| DOC (± SE) (µM) | 483.11 (±68.40) | 297.36 (±3.08) | 474.56 | 318.44 (±9.42) |
| DON % of TDN | 67.03 | 76.58 | 95.48 | 85.33 |
| Temperature (ºC) | 3 | 4 | 18 | 7 |
| Salinity | 6.30 | 6.10 | 6.3 | 7.3 |
| Chlorophyll a (µg/l) | 0.30 (±0.00) | 2.34 (±0.27) | 6.49 (±0.01) | 1.76 (±0.04) |
| C/N ratio | 42.23 | 23.68 | 30.11 | 17.78 |


Table 2. Wastewater effluent nutrient content for the different seasons sampled. C:N
ratio is calculated as the ratio DOC:DON (moles).

| | Winter | Spring | Summer | Autumn |
|---|---|---|---|---|
| Date | 23/01/2013 | 03/04/2013 | 16/07/2013 | 25/10/2013 |
| TDN (± SE) (µM) | 600.12 (±6.56) | 576.20 (±3.20) | 518.39 (±2.39) | 498.20 (±9.77) |
| $NO_2^-$ (± SE) (µM) | 8.00 | 32.74 | 29.44 (±0.04) | 29.29 |
| $NO_3^-$ (± SE) (µM) | 81.00 | 113.64 (±2.17) | 192.00 (±6.38) | 228.57 |
| $NH_4^+$ (± SE) (µM) | 7.76 | | 117.93 (±1.20) | 165.15 (±1.21) |
| $PO_4^{3-}$ (± SE) (µM) | 0.02 | | | 0.19 |

| | | | | |
|---|---|---|---|---|
| DON (± SE) (μM) | 503.35 (±2.93) | 429.83* | 179.02 (±7.95) | 75.20 (±4.39) |
| DPA (± SE) (μM) | | 18.71 (±2.64) | 2.64 (±0.17) | |
| DOC (± SE) (μM) | 1347.96 (±205.65) | 924.18 (±6.66) | 1082.37(±2.50) | 706.87 (±9.99) |
| DON % of TDN | 83.88 | 74.60* | 34.53 | 15.09 |
| C/N ratio | 2.68 | 2.15 | 6.05 | 9.40 |

*Calucated without $NH_4^+$ concentration (overestimation)


Table 3. Statistics for the fitted models for the different metabolic rates and the
variables that explain its variability, to account for pseudo-replication incubation day
nested to season (i.e. experiment) was included as random factor. p was calculated
comparing nested models. SE: standard error; N: number of observations.

| | Estimate | SE | t Ratio | p | $R^2$ | N |
|---|---|---|---|---|---|---|
| GPP | | | | | 0.84 | 73 |
| Intercept | 27.71 | 5.45 | 5.09 | | | |
| DOC (μM) | -0. 007 | 0.007 | -0.97 | < 0.0001 | | |
| | | | | | | |
| CR | | | | | 0.84 | 73 |
| Intercept | 23.02 | 3.37 | 6.83 | | | |
| DOC (μM) | -0.006 | 0.005 | -1.38 | < 0.0001 | | |
| | | | | | | |
| NCP | | | | | 0.79 | 77 |
| Intercept | 4.85 | 2.68 | 1.81 | | | |
| DOC (μM) | -0.002 | 0.004 | -0.41 | < 0.0001 | | |
| | | | | | | |
| BP | | | | | 0.91 | 92 |
| Intercept | 1.11 | 0.45 | 2.47 | | | |
| DOC (μM) | 0.001 | 0.001 | 1.30 | <0.0001 | | |
| Nitrate (μM) | 0.02 | 0.004 | 5.17 | <0.0001 | | |
| Phosphate (μM) | -1.00 | 0.32 | -3.12 | <0.003 | | |
| DON (μM) | 0.02 | 0.01 | 2.19 | <0.03 | | |


Table 4. Results of MANTEL tests (Pearson's r) to examine if absolute shifts in
bacterioplankton community composition were correlated to absolute changes specific
environmental variables and metabolic rates measured in the incubations during the
experiments. Significance is indicated in parenthesis.

| | **All** | **Winter** | **Spring** | **Summer** | **Autumn** |
|---|---|---|---|---|---|
| Date | - | 23/01/2013 | 03/04/2013 | 18/07/2013 | 04/11/2013 |
| Temperature | 0.5118 (0.001*) | 0.1481 (0.299) | 0.208 (0.123) | 0.1582 (0.558) | -0.01759 (0.489) |
| NCP | 0.05345 (0.149) | -0.2466 (0.689) | 0.2233 (0.089) | 0.05968 (0.242) | -0.06 (0.573) |
| GPP | 0.2095 (0.004*) | -0.2182 (0.591) | -0.1855 (0.795) | 0.1588 (0.09) | 0.08498 (0.277) |
| CR | 0.2651 (0.001*) | -0.4532 (0.862) | -0.211 (0.874) | 0.2085 (0.044*) | 0.385 (0.014*) |
| BP | 0.3208 (0.001*) | -0.1194 (0.627) | 0.3048 (0.047*) | -0.04983 (0.658) | 0.1228 (0.218) |
| Chl $a$ | 0.2147 (0.001*) | 0.1021 (0.396) | 0.1326 (0.178) | 0.3575 (0.005*) | 0.02732 (0.398) |
| DOC | 0.03036 (0.272) | -0.1072 (0.600) | 0.1926 (0.134) | 0.269 (0.035*) | 0.04995 (0.357) |
| TDN | 0.1558 (0.003*) | -0.03911 (0.513) | -0.04881 (0.497) | 0.247 (0.027*) | 0.04071 (0.321) |
| $NO_2^-$ | 0.1558 (0.003*) | -0.03979 (0.531) | -0.04449 (0.683) | 0.01229 (0.376) | 0.1027 (0.181) |
| $NO_3^-$ | 0.05622 (0.111) | -0.01186 (0.457) | -0.06687 (0.65) | 0.03073 (0.328) | 0.1416 (0.161) |
| $NH_4$ | 0.02908 (0.311) | 0.00467 (0.361) | -0.08367 (0.611) | -0.00490 (0.433) | 0.1069 (0.195) |
| DON | 0.00043 (0.391) | -0.09584 (0.667) | -0.04767 (0.452) | 0.136 (0.163) | 0.03776 (0.356) |
| DPA | -0.01335 (0.529) | -0.03385 (0.49) | -0.1055 (0.612) | -0.00163 (0.407) | -0.03274 (0.532) |
| $PO_4^{3-}$ | 0.2982 (0.001*) | 0.1492 (0.207) | ND | 0.2853 (0.007*) | -0.1585 (0.819) |



**Figures captions**
Figure 1. Chlorophyll a content for the different incubation days and different
treatments for the four experiments.
Figure 2. Gross primary production (GPP) in mmol $O_2$ $m^{-3}$ $d^{-1}$ measured the seven
incubation days for the different treatments and experiments.
Figure 3. Comparison of actual values and values predicted by the mixed effects
model for (a) gross primary production (GPP), (b) community respiration (CR), (c)
net community production (NCP) and (d) bacterial diversity. Black solid line
represents the 1:1 line.
Figure 4. Community respiration (CR) in mmol $O_2$ $m^{-3}$ $d^{-1}$ measured the seven
incubation days for the different treatments and experiments.
Figure 5. Net community production (NCP) in mmol $O_2$ $m^{-3}$ $d^{-1}$ measured the seven
incubation days for the different treatments and experiments.
Figure 6. Bacterial production in $\mu$g C $L^{-1}$ $h^{-1}$ for the different measured days for the
different treatments and experiments.
Figure 7. Differences in alpha-diversity, estimated from Shannon index, between
controls and nutrient amendment, i.e. all nutrient amended treatments were binned
and compared against all controls. Circles denote variation in alpha-diversity within
the binned samples where colour corresponds to different treatments.
Figure 8. Relative abundances (i.e. percentage of total sequences) of major bacterial
groups at phyla/class level in the different treatments and experiments. Colour denote
specific groups.
Figure 9. Correlations between shifts in relative abundances of major bacterial groups
at phyla/class level and environmental factors and metabolic activity. The level of
correlation is estimated from Pearson r where blue and red colour indicate negative
and positive correlations, respectively.

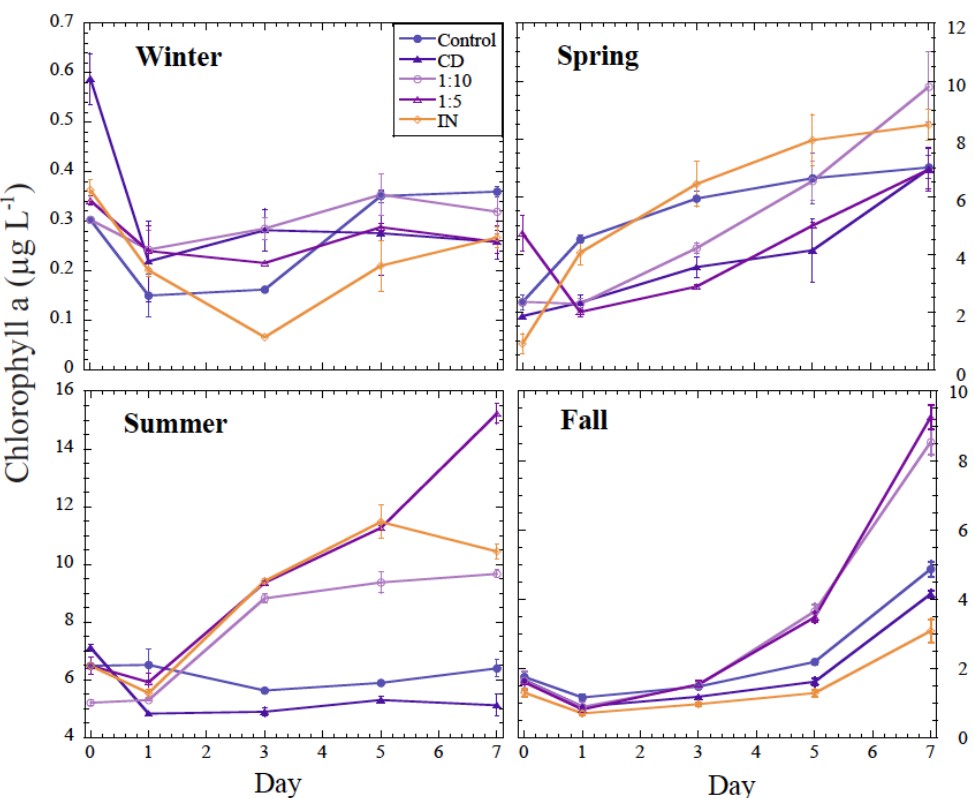



Figure 1



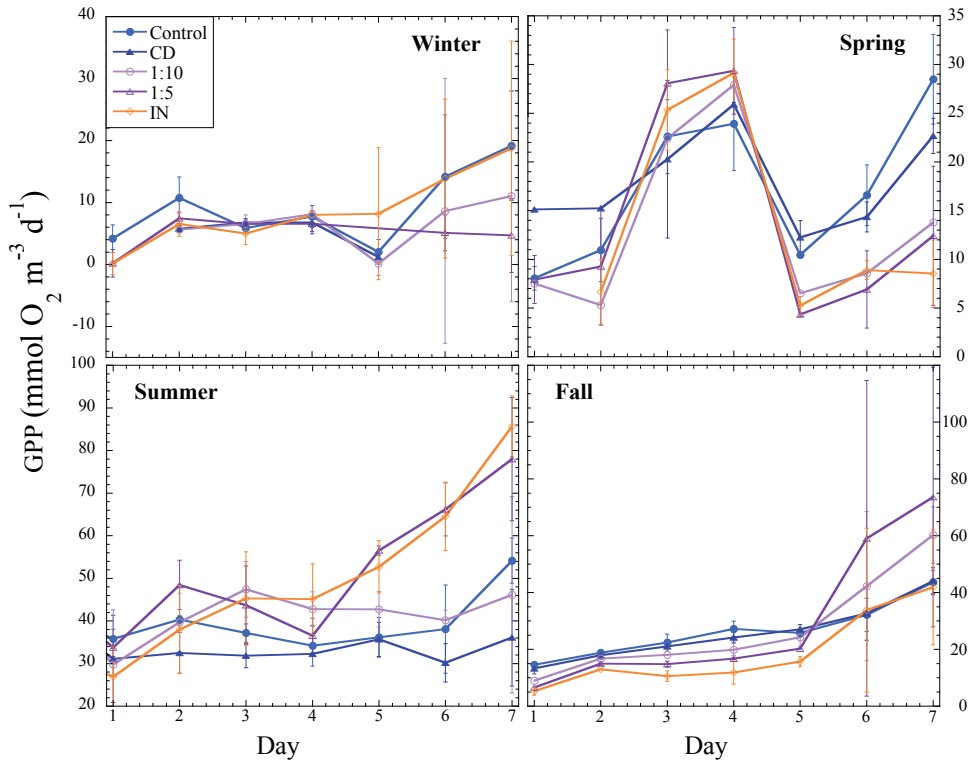


Figure 2

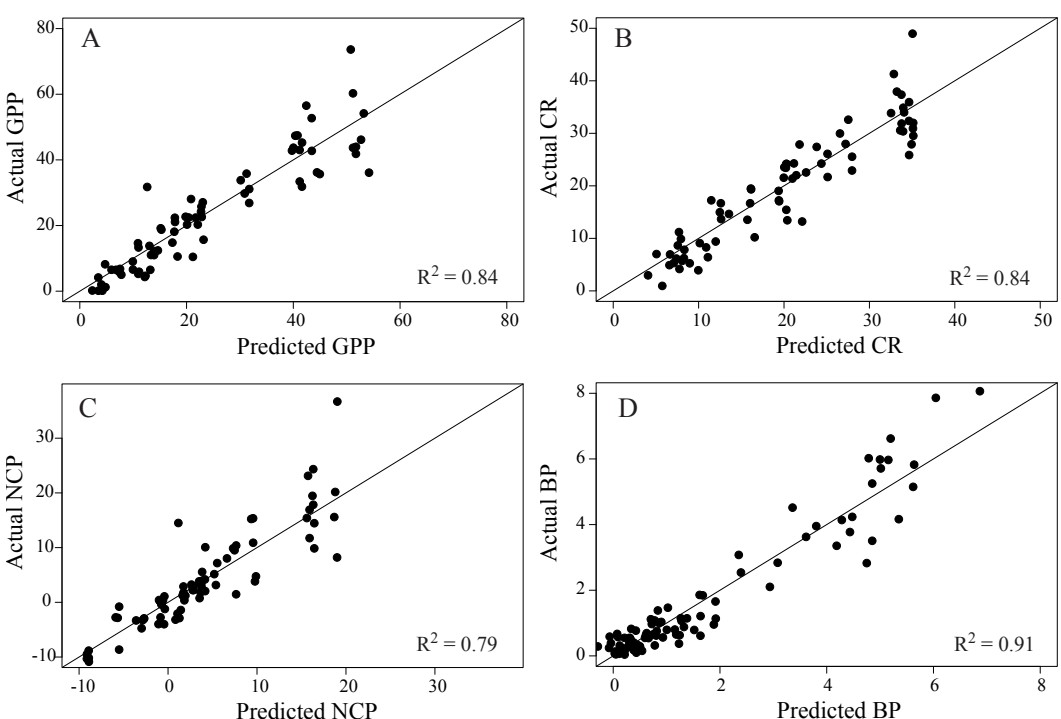


Figure 3

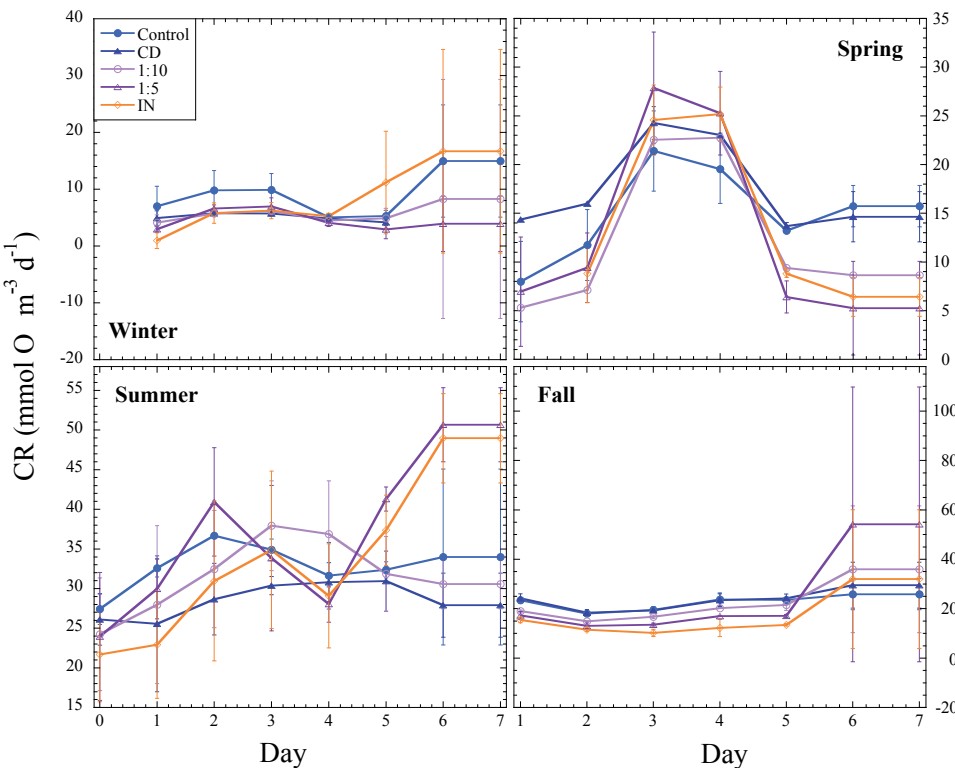

Figure 4

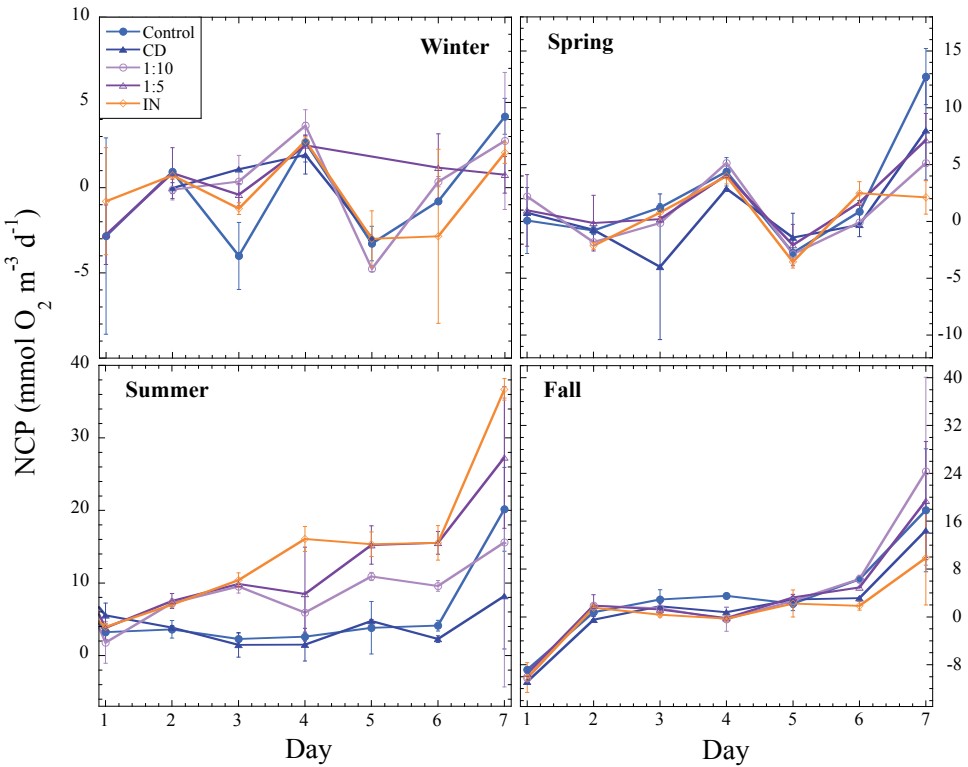


Figure 5



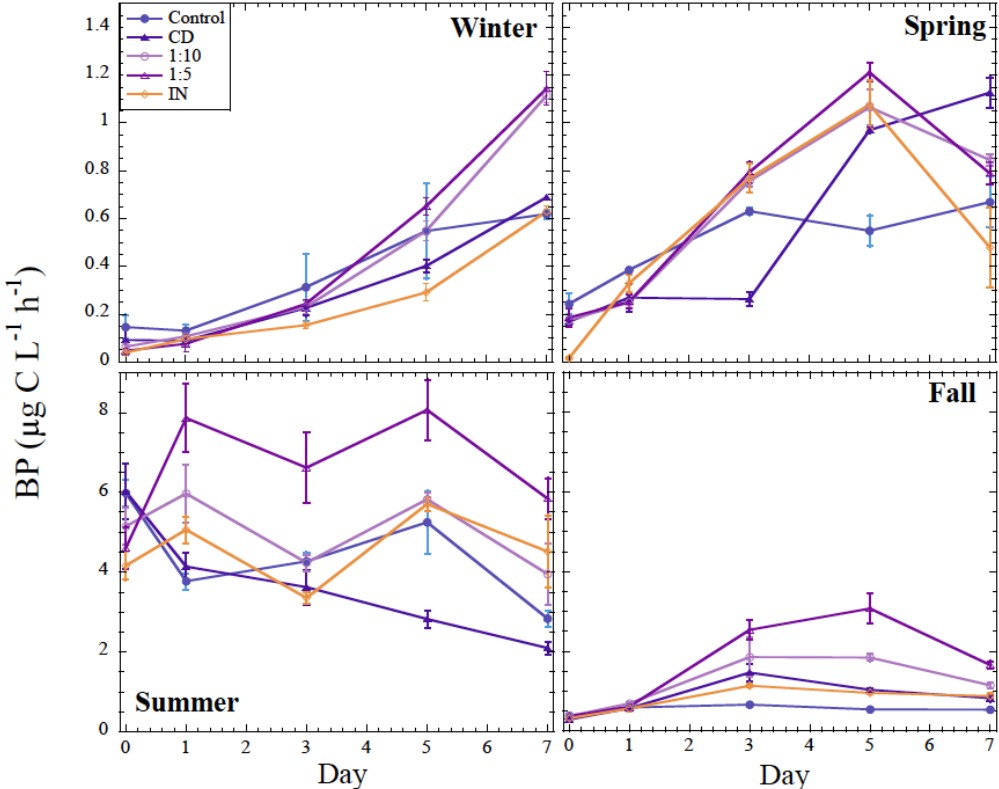


Figure 6


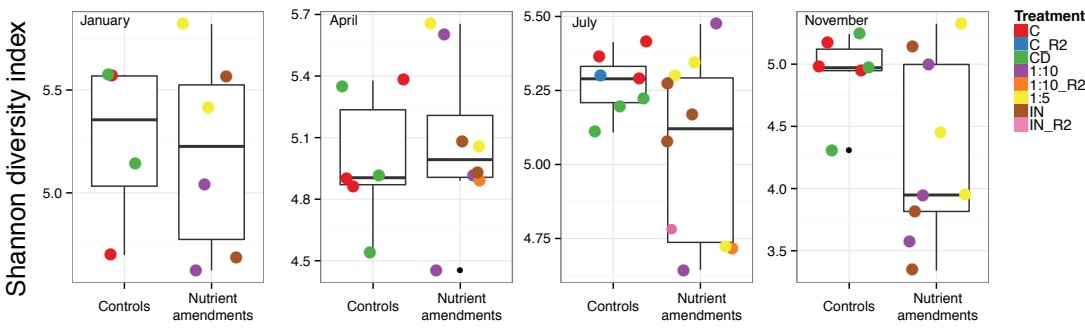

Figure 7

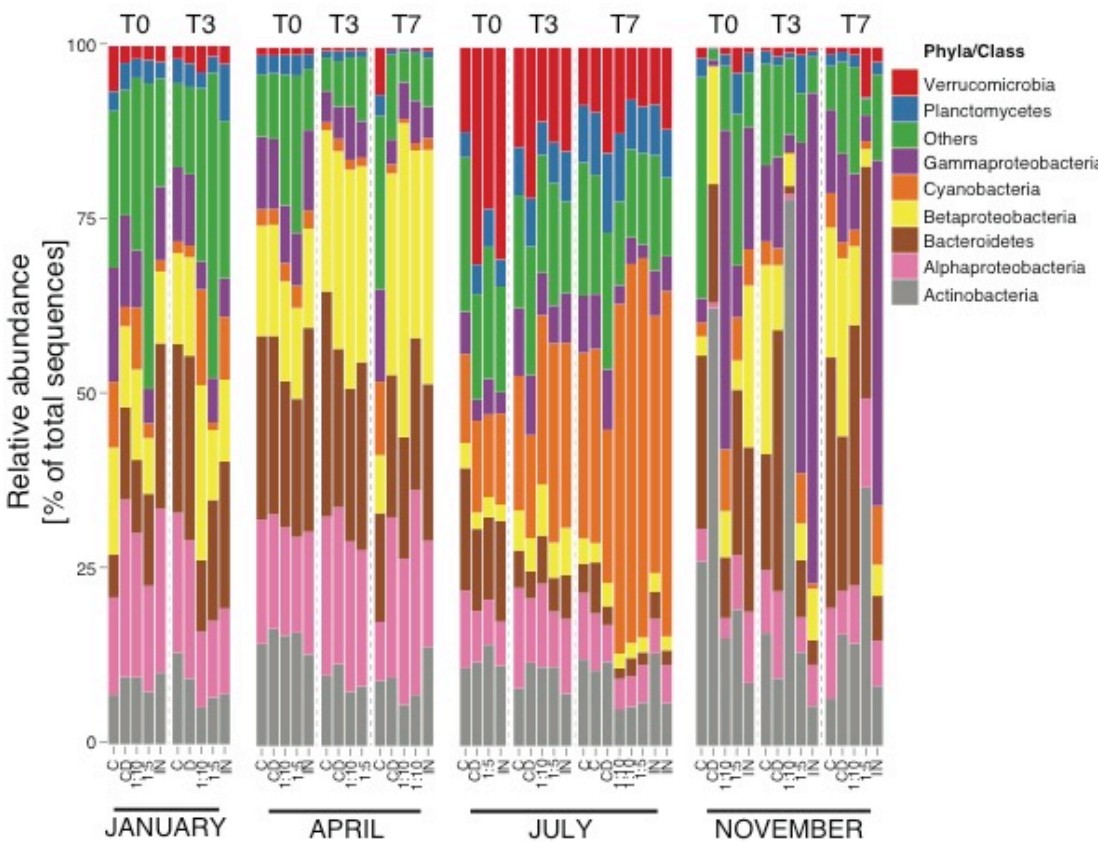

Figure 8

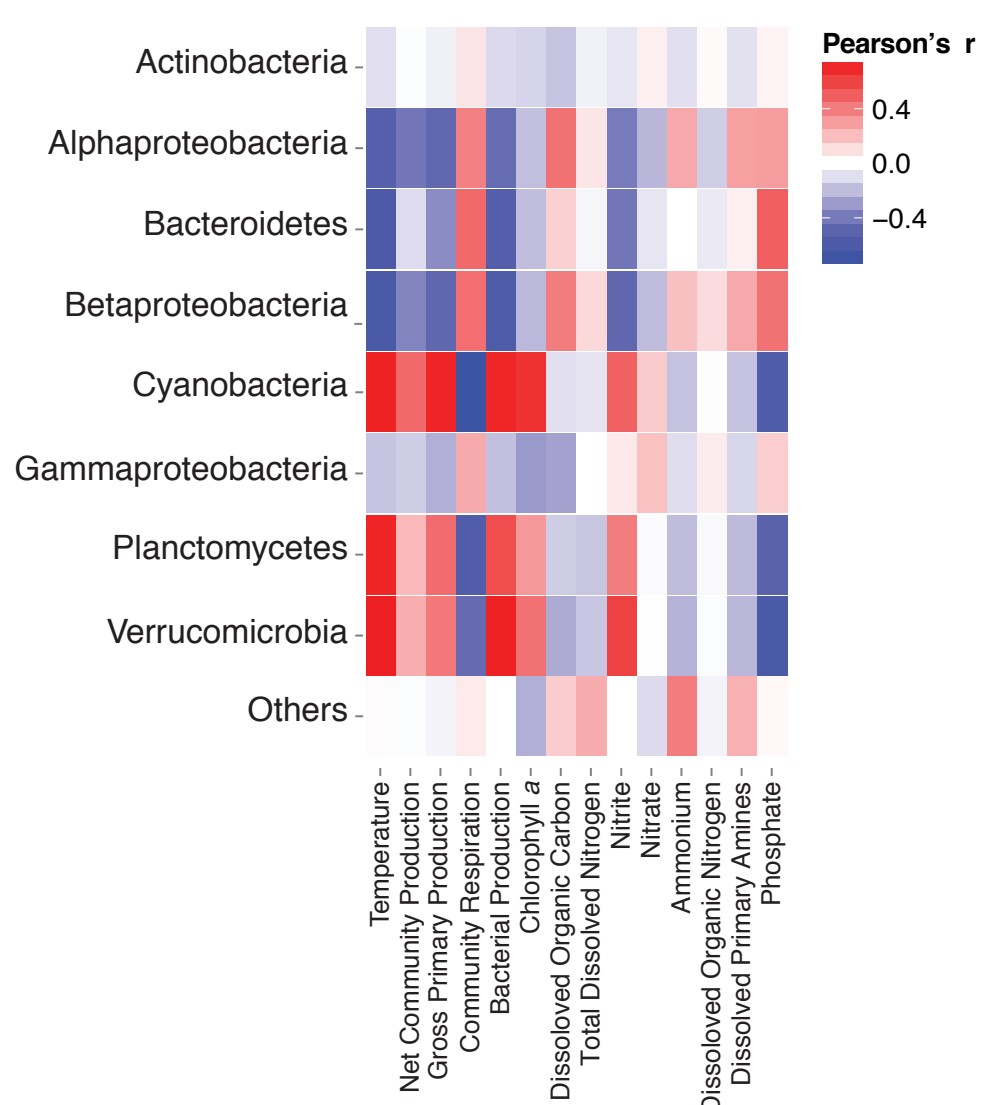


Figure 9