# Peer review of "Effects of wastewater treatment plant effluent inputs on planktonic metabolic rates and microbial community composition in the Baltic Sea."

_Biogeosciences, 2016_

## Referee Comment (RC1) · Anonymous Referee #1 · 25 May 2016

Comments

The authors tested the effect of the wastewater treatment plant (WWTP) effluent inputs on Baltic Sea planktonic communities in 4 experiments. They did so sampling seawater during winter, spring, summer and autumn and observing the effect of different WWTP addition to natural communities. They observed that nitrogen-rich DOM inputs increased BP and decreased PP. This trend will drive to an increase of carbon consumption and shift the ecosystem toward heterotrophy. Although the experiment was well performed and that the authors analysed several variables, I had the feeling that

the paper was written in a hurry, sometimes with lack of precision and details. However, the study and the results obtained here are strongly recommended to be available for scientific community. Then, I recommend this manuscript to be published with minor revisions.

Detailed comments: Introduction 65: For non-specialists in WWTP, I think that you should explain why you are talking about the Chesapeake Bay and its discharge limit. Is it a bay that receives important WWTP effluent at the US scale? Comparable with the study area? 58-73: You should explain a little bit more how is the WWTP effluent in the Baltic Sea. What is its average discharge? TN, DIM, DON, DOM concentrations? Do any treatment have been implemented in order to reduce the WWTP discharges into the Baltic Sea?

Methods 87-90: Why did you choose to sample during the four seasons? Does the effluent discharge more during winter due to rainfall and enhance the WWTP inputs on Baltic Sea? 93-96: Did you check your WWTP after being filtered and frozen if some organism remains into your sample (as some bacteria may be smaller than 0.2 $\mu$m) using for example flow cytometry? 102-103: In my opinion, the third treatment is unclear. 103-105: I am not sure that I understood how did you do CD. You said that CD consisting of seawater diluted with milli-Q to have the same portion of community that 1:10, 1:15 and IN. Does that mean that you made the same "dilution" than the WWTP treatment but instead of using WWTP, you used milliQ and you made a CD of 1:10 (100mL of milliq for 1000ml of seawater), another of 1:5 (200ml of milliq for 1000ml of seawater) and another as IN (?)? If this is the way you made, I don't understand how do you have just one CD. . . 122-137: In this paragraph, you didn't explain how did you do dark and light incubation, how many replicates did you have. You just said "bottles were incubated at the in situ temperature. . . during one week". Confinement methods prone to error as you might exclude zooplankton, enhance trophic interactions within the bottle and so on. Most of the incubations realized to estimate changes in DO in order to determine NCP and CR lasts between 12-48h. One week of incubation is a

lot. Do you think that the community inside your bottles after 3-4 days of incubation is representative of in situ community that may receive once an amount of DON or DIN 3-4 days earlier? You did not talk in this paragraph about the confinement effect and the effect of 7-days incubation. I think that it is really important and that you should take time to write about it and give some criticism of your estimates. 130-133: You explained that incubations were illuminated by artificial light with a mean PAR of 1373.2 uW/cm2. Why did you choose this amount of light? Is it representative of the daily PAR that Baltic community receives at 2m depth? Does the illumination was constant during light exposition or does it increase until reaching a maximum natural irradiance and then decrease? The light hours range for the summer experiment is about the double of light for winter experiment. Do you think that GPP is comparable in summer and in winter experiment? Would it not be better to express GPP per hour and not per day? 134-137: I understood that you estimated NCP this way: DOmin1-DOmin0, DOmin2-DOmin1, DOmin3-DOmin2, ... DOmin1440-DOmin1439. Then, you had NCPmin1, NCPmin2, NCPmin3,..., NCPmin1440 and you sum it to have it per day. Is it right? Or did you make it directly DOmin1440-DOmin0=NCP24h? According to the calculation that you made, did you compare it with the other way? Is it similar? 141: Can you explain what is the killed control please? 140-142: It is not specified during the 60min incubation the temperature and irradiance received by the samples.

Results 211-245: In the Metabolic rates results section, I missed to read more about statistics. For example, does GPP was significantly different between each treatment for each season? The same for CR and NCP. 223: I think you wanted to write "GPP decreased with Chl a content, it increased with DOC concentration" in reference with Table 3.

Discussion 345-348: I don't agree with this part of your discussion. I really don't see from which results you concluded that DOM significantly increased BP and decreased NCP, GPP and CR. Where are the statistical tests showing that GPP, NCP and CR were significantly different from control and CD? I understand that in BP results (247-260),

you observed a tendency to increase with higher addition of effluent (nothing significant I guess), that you observed significant differences in BP for different sampling day, treatments, interaction between sampling day and treatment but there is no mention in this paragraph that BP was significantly higher at the end of the experiment with effluent addition. . . Furthermore you observed no significant differences between treatments for spring and winter experiments. I don't see in this results section any mention of "BP was significantly higher under effluent addition than control and CD at the end of the experiment" For GPP, NCP and CR, you even didn't show any statistical test in the results that can lead you to this conclusion. In Figs. 2, 4 and 5, we can see that the responses are different according to the season, that responses aren't linear along the experiment (response different at day 3 than at day 6) but you didn't talk about that in the result section and we missed that. Looking to these figures, I don't agree with the general conclusion that NCP, GPP and CR were suppressed by DOM addition, but if some statistical tests show me the inverse ok. . . but I need to see it! And, did you consider that the "good" response is at day 7? If it is so, why? Why not at day 3? 354: recent references? 359: Correct "deceased" by "decreased" 358-360: Did BGE increase and decrease significantly? R2, p? If not, you should specify it too. 360-363: The example that you presented from the Bothnian Bay didn't seem really significant. . . Can you find other references showing significant increase of BGE with nutrient addition? 368-371, 381-393: Again, I disagree with the conclusion of a reduction of PP with effluent addition and that planktonic community in this region will shift toward heterotrophy. Either you have to improve your results showing statistical tests that can insure your conclusion that in general planktonic metabolism decreased under effluent addition or remove it. 432-434: There are more references about it and you should add few of them. . . not only yours âŸž 436: Use the same bibliography style than previously (Vaquer-Sunyer et al. 2015)

Conclusion 441: "DOM from WWTP effluent is nitrogen-rich." Remove this sentence. 442-461: Same thing than previously, I disagree with your conclusion.

Table 1 Nutrients formulas should be written correctly with subscript and superscript. Add C/N ratio to know if sampled communities were N limited or not.

Table 2 Idem (nutrient formulas)

Table 3 R2 not R2

Figs, 1, 2, 4, 5, 6 For each figure, it could be better to have the four season plots with the same axis range in order to compare the seasonal variation of each variable.

---

## Referee Comment (RC2) · Anonymous Referee #2 · 31 May 2016

The study by Vaquer-Sunyer describes the effects of wastewater treatment plants effluent inputs on different microbial parameters. I find the topic interesting and relevant for publication in BG. However, the approaches used are not well explained and therefore it is hard to judge the results. Moreover, I miss precise definitions and statements in the discussion. Therefore the manuscript seems in the current status premature.

Introduction Line 50 ff. The introduction (and in the discussion) gives a very anthropogenic view about hypoxia. Various prokaryotic groups' life in oxygen minimum zones and often exists only at those conditions (read for example "Microbial ecology of expanding oxygen minimum zones" Jody J. Wright, Kishori M. Konwar & Steven J. Hallam). From a microbial point of view the biological diversity therefore increases by the presence of oxygen minimum zones. Moreover, oxygen minimum zones in the central Baltic Sea are connected to the presence of a halocline that prevents water mixing in spring and autumn and therefore a typical phenomenon in the Baltic Sea. The authors mean coastal water hypoxia (as in the cited Conley 2011 publication) that should become clearer in the introduction (and discussion).

Methods:

Line 105-105: what salt solution was used?

Line 131-133. Use space between unit and number.

Line 136: please give a short description who the calculation of the metabolic rates were performed. If I understand correctly, CR was estimated from change in oxygen over night and GPP from the sum of NCP and CR. Since DOC is produced during daytime the CR can differ between night and day. Typically incubations are performed at light and dark conditions in parallel to estimate CR and GPP. Please discuss that your approach gives comparable results as the parallel incubations.

Line 151: The 341-805r primers were designed for bacteria the protocols in Hugerth et al are for eukaryotes. If I understand correctly a two step PCR were performed, please describe the protocol in more detail, especially if the PCR contained several independent PCR cycles. This is important information since this can introduce a strong bias into the abundances estimates for bacteria.

Line 119: What happened to the data of the DPA measurements?

Result

Line 190 221 table 1 and 2: How was DOC and TOC measured (replicates, sensitivity,. . .)?

Line 273: How was the change in the number of OTUs?

Discussion

Line 345: It is unclear what the abbreviation DOM stands for. Typically it is used for dissolved organic matter (which was not measured in this study), but in the manuscript it reads rather that N-rich dissolved organic matter is meant. Please clarify. The results suggest that certain concentrations of nutrients (Figure 2; figure 4, Figure 5 1:5; IN) cause a change in the GPP, CRR and BP compared to the control treatment. Moreover the statistics give often DOC and CHla as the variances explaining the variability (line 224, line 234, line2 44) and only for bacterial production a correlation with NO3. Therefore the statement: " OM significantly increased bacterial production, whereas it decreased gross and net primary production and community respiration rates" seems not justified.

Line 413. Alpha diversity is not only expressed in the Shannon index. I think you mean Shannon index here, but it would be interesting to see how the total number of OTUs (richness) change.

Please discuss also the artifacts that can arise from long term bottle incubations especially for t7.

P has also a strong impact on the eutrophication in the Baltic Sea since many Cyanobacteria are able to fix N (read for example "Andersson, A., Höglander, H., Karlsson, C., and Huseby, S. (2015). Key role of phosphorus and nitrogen in regulating cyanobacterial community composition in the northern Baltic Sea. Estuar. Coast. Shelf Sci. 164, 161–171."). Since P was also measured in the experiments, and found to have a strong influence on the BCC (Line 271), I wonder why P is rarely discussed.

Figures

The legend of figure 8 is almost not readable

The colors in the plot are not easy to distinguish the abbreviations should also be given

in the legend

Figure 2 and Figure 4: they are based on the same data, whereas one contains subtracted CR? Therefore I think one of them is sufficient.

---

## Referee Comment (RC3) · Anonymous Referee #3 · 1 Jun 2016

General Comments:

The current study by Vaquer-Sunyer and colleagues describes the effects of wastewater treatment plant effluent inputs to the Baltic Sea on coastal planktonic microbial communities. The topic should be of interest to readers of Biogeosciences, and should be able to be made appropriate for publication after revision. The authors tested a number of relevant microbial parameters, and the experiments appear to be well-thought-out and executed, although some of the methods require some clarification. The main results showed an increase in bacterial production and decreases in primary production and community respiration following amendment with wastewater, along with some changes in bacterial community composition. There is some confusion, especially during the discussion section, between discussion of metabolic activity vs. community composition - i.e. it seems that an increase in BP and decrease in PP is taken to indicate a shift in community type (autotrophic to heterotrophic), which was not tested or substantiated by the data in the paper. I find that the discussion section in particular becomes somewhat disjointed, and that some of the conclusions drawn from the study are overstated (i.e. presented with more certainty than the data allow). As such, the paper requires more thought and more careful presentation before it is ready for publication. I hope that specific comments below are helpful in this regard.

Specific Comments:

Introduction: Line 67-69: This statement regarding reduction of TN seems quite specific. Can a reference be provided?

Methods: Line 127: "in situ temperature" - are these the temperatures listed in Table 1? If so, please refer to Table 1 here.

Line 136: A description of the method to calculate metabolic rates (even if it is an abbreviated summary) should be provided here, not simply a reference to another paper that describes the method. I looked up the other paper, and it is not clear to me how all of the metabolic rates were derived from the data in the current paper.

Line 136: "water properties" - please either list all of the properties (maybe a better term would "physicochemical parameters") used in the statistical models, or refer to the table that contains them.

Line 154: What are the "biological replicates"? I did not get this from the description of the experimental/treatment design. Given this, I think that the treatment description (Line 97 onwards) needs to be improved - I see four experiments (one for each season) with five treatments each, but no replicates. Perhaps a list of everything that was tested

for each treatment within each experiment should be included. It's not clear to me what exactly was measured on which sample.

Results: Line 186: Can you please clarify whether the nutrient determinations were done on the samples collected for each expieriment, following the filtration and freezing steps described in the Methods? I suggest making this clarification in the Methods so that the reader knows exactly where the reported data are coming from.

Line 192: The Methods section should be updated to include how the seawater samples for nutrient and chlorophyll analysis were collected and handled. I see a description of planktonic microbial community sample collection only. The description of how the samples were analyzed for nutrient and chl content, is complete, just not collection, filtration, storage, etc.

Line 203: "as a consequence of re-mineralization" is probably a good assumption to describe increasing nutrients, but because the source of the increased nutrients was not tested in the current study, this statement (and its degree of certainty) is not appropriate for the Results section.

Lines 204-209: I do not see a description in the "statistical methods" section of the Methods that could have been used to arrive at these conclusions regarding cal. The methods seem to cover metabolic rates and community structure, but no the relationships among physicochemical variables such as Chl and light. Please clarify this in the Methods.

Line 255: Rather than saying that BP 'depended on' DOC, it might be more useful to describe the direction of the relationship.

Line 267: "temperature significantly explained..." I question whether strong conclusions can be drawn regarding the influence of temperature. Given the range of temperatures (3,4,7, 18), it seems that the single high temperature (18) is an outlier and would exert extra influence on the correlations in the Mantel test. Can you address this in this

review response, since many of the relationships in the paper seem to revolve around temperature?

Line 273: "relatively similar" is unclear. Perhaps provide the range in alpha diversity across all experiments in parentheses and say "similar".

Line 275-276: Can you clarify the wording please? I think what is meant is "a lower Shannon index was observed for all nutrient treatments compared to the controls", but I am not certain based on current wording.

Line 281-282: Is the implication here that the Betaproteobacteria decreased in the control over time, rather than increasing in the treatments? That is interesting, and I suggest making clear what the conclusion related to this result is. Also, if there were changes in the control during the experiment, is there concern over bottle effects?

Line 284-285: "higher relative abundance" - Can you please add in parethenses how much higher the relative abundance was, compared to other treatments and controls (on average)? Also, is there statistical significance associated with this statement? It is fine if there is not, but I still suggest providing some numbers so that the reader can make the comparison more clearly.

Line 290: "increased in the control" Same question as above - With so many changes in the control, are we just seeing bottle effects over time? Can you comment on the validity of comparing these long incubations? Why would things be changing in the control?

Line 301: Can you please define what is meant by "finer phylogenetic scales"? i.e. at OTU level? Phylum level?

Line 301: "when communities responded to experimental treatments" I'm not sure what this means. Can you clarify whether you mean that you only looked at links between environmental and biological factors in experiments where there was a response to the treatment? Perhaps this needs to be split into more than one sentence to make the

meaning clearer.

Line 302: "were positively correlated" Is this referring to the relative abundances of these groups? Can you please say what about these groups was correlated with temperature?

Line 309: It is not clear to me where the explanation of the variance is coming from here. Earlier in the paragraph, Pearson correlation is referred to, but I am not sure that makes sense here. Can you please specify?

Line 311: "8 major phyla" Are these 8 major phyla/classes listed somewhere in the paper? If not, please do so here.

Line 320: "strong correlation" Can you please say in parentheses what constitutes a "strong correlation"?

Line 321: Why "e.g."? Can you list all of the strong correlations, or only these few because there are too many?

Line 325: What is a substantial correlation? Please give a range, or an average, especially since the data is in the supplement. Listing something here allows the reader to better understand the relationship.

Discussion: Line 356: The type of modeling exercise described in this section is valuable, and can be used to support a hypothesis, but I would caution against using the term "validate" in this case. It implies a level of certainty that I do not think can be reached in the current study.

Line 359-360: Could you please provide the coefficients for each parameter in this model, so that the reader can get an idea of the rate of change in BGE associated with each variable? They could be listed as a rate in parentheses after each parameter, for example. Were all of the parameters "significant" in the model? How was the model selected?

Line 368: Bacterial carbon demand was not measured in this study, rather the authors assume it based on community respiration. This statement should be amended to reflect the level of certainty that can be supported by the data.

Line 369: The reduction in primary production does not lead to more carbon being used by the microbial loop. More carbon is used by the microbial loop because bacterial production (or respiration, which was not measured) increases.

Line 371: Could you provide a min-max range of the ratio of BP:NCP from your experiments to support this point (that the ecosystem moves towards heterotrophy)?

Line 372: Increasing carbon flow into the microbial loop should not result in reduction in the transfer of carbon to higher trophic levels. Organic matter entering the microbial loop through bacterial uptake should still be returned to higher trophic levels through coupling with the traditional food chain. The paper by Wohlers refers to a decrease in carbon fixed by primary production being transferred to higher trophic levels (not organic C uptake by heterotrophs), and (as far as I can tell), the Berglund paper simply suggests that increased runoff (and thereby nutrient inputs) and temperature should favor a heterotrophic bacteria based food web and decrease production. Either way, I can't see why the authors conclude that increasing carbon flow into the microbial loop alone should result in a reduction in C transfer to higher trophic levels.

Line 379-380: It's not clear to me how this is related to the current study or discussion.

Line 381-382: "A change in the planktonic community towards more heterotrophic communities" So far, this discussion has pointed out that rates of BP increased and the NCP decreased, with the addition of DOM. However, I don't think that there is evidence here that the community composition is shifting towards heterotrophy? Or, if there is evidence of this, it should be mentioned in the discussion here before lines 381-382.

Line 382: While it is true that a reduction in photosynthetic rates would decrease oxygen production, I do not see clear evidence from this study that a shift towards

heterotrophic communities is occurring, or that any reduction in photosynthetic rates would be the result of such a shift. In short, Line 381-382 make some assumptions that should be revisited and substantiated with data, if it exists. If it does not, then this discussion point should be reworded so that it is supported by the data.

Line 390: "reducing the ecosystem capacity of removing nitrogen" Doesn't anoxia favor the removal of nitrogen (i.e. denitrification)?

Line 390 - 393: The final two sentences here (lines 390-393) do not flow from the previous discussion about anoxia and eutrophication. The text in this section should be revised to make clear points and conclusions, which are supported by the data.

Line 399: "disturbances" - Do you mean effluent inputs? what is meant by "disturbances"?

Line 401: Can temperature be de-convoluted from season or other parameters? The changes in temperature weren't really "experimental" changes, but matched the in situ conditions at the time of sample collection, correct? I guess I don't really understand what is meant by "changes in temperature". And I still have the concerns listed above regarding the range of temperatures and influence of outlying temperature.

Line 409 - 412: The sentence beginning "It is noteworthy" is not clear. The authors are suggesting that what changes in what relationships? Changes in the relationship between composition and function? What is the relationship between composition and function? I'm not sure where this sentence is going.

Line 410: the link between community composition and function should be susbtantiated with a reference.

Line 411-412: This is redundant with the discussion of theoretical BGE above.

Line 412-413: Lower diversity doesn't necessarily equate loss of function - aren't many functions redundant within a microbial community?

Line 422: "caused responses" What responses? It would be more correct to say that these certain populations responded to effluent inputs. Also, didn't the Verrucomicrobia increase in the control? So how are changes in Verrucomicrobia associated with effluent inputs?

Line 424: Nutrient inputs, or effluent inputs? The terminology used here is confusing, and I can't tell exactly what the authors are trying to conclude.

Line 430-431: "warming could increase cyanobacterial blooms" How did cyanos in this study respond to temperature?

Line 434-438: As written, this closing sentence (which should be used to drive home a major point of the current study) seems to focus on the results of a previous study instead. How does the current study and its findings support or add to the findings from the previous study? Also, the finding that "warming and effluent inputs increased planktonic respiration and bacterial production faster than primary production" is attributed to the previous study - I thought that this was a new conclusion of the current study? If not this, then what IS the new conclusion of the current study?

Line 443: The conclusion that this leads to an increase in BGE is stated with more certainty than can be derived from the current study. It assumes that the decrease in CR is also a decrease in BR, but that may not be the case. The conjecture is ok, but should not be stated as fact.

Line 448-449: If cyanos increased in summer, how is this be linked to effluent inputs and not temperature? Also, I assume that "abundance" is "relative abundance"?

Line 454: If cyanos are increasing due to effluent input, it is not clear to me how the conclusion that planktonic communities are shifting toward heterotrophic communities is made? Were the relative abundances of photo and heterotrophic organisms compared? Or is this based on rates of activity of the two groups? If the latter, this should be rephrased so that it does not lead the reader to conclude that the community structure is changing, and is responsible for a shift towards heterotrophy.

Line 460: Low BR (not CR) compared to BP leads to high BGE. Since BR was not measured in this study, the authors should be careful regarding the level of certainty they assign to these conclusions. While interesting, any conclusion related to BGE is theoretical and should be used to guide further research, not stated as fact.

Tables and Figures:

Table 1: As the table contains more information than only nutrient content, a more descriptive caption should be used. Perhaps "physicochemical parameters" would be more appropriate. The caption should also reflect the number of replicates used to arrive at the listed standard errors. Chemical symbols for nutrients should be listed with proper superscripts, subscripts, and charges (throughout the text as well). If all other chemical species are listed in molar concentrations, DOC should be too. It is best to keep these consistent.

Table 2: I notice here that the amount of P added is unknown for half of the treatments, which makes me question results related to changes in P. Can the authors address this please? Why is the carbon labeled as "TOC" if the samples were filtered as described in the Methods? How was C:N ratio calculated? Is it a ratio of DOC:DON? or DOC:DIN? Is it by mass, or moles?

Table 3: Some explanation of all factors tested and the model selection parameters should appear in the Methods section. How was the best model chosen? Were all factors tested initially (and what are all the potential factors)? The random factor for "experiment" is referred to as "season" in the Methods section, is it not? Please change one or the other so it is consistent.

Table 4: I notice a lack of correlation with organics - does this not imply that the shifts in composition are not related to effluent? "specific environmental variables" Which environmental variables were used? It seems that there should exist a table, similar

to tables 1 and 2, that gives the environmental parameters for each incubation (Tables 1 and 2 show environmental parameters at the collection site and in the effluent, respectively, correct?).

Figures 2, 4, and 5 should be of higher resolution. They appear blurry in the pdf.

Figure 3: What is a whole model plot? What is the model? Please clarify this caption. And if whole model refers to some sort of model selection, it should be described.

Figure 7: "A" is not labeled. It doesn't seem to be possible to see all of the treatments in figure A. I think only showing Figure B would be more informative.

Figure 8: It would be easier to look at if we could see the controls first in the group for each time point and if the time points were separated somehow, perhaps by a small line

Figure 9: What were the highest and lowest correlations? Cutting it off at 0.4 seems like it would bin together a lot of data, unless there are no strong correlations. If there are no correlations >0.4, this should be made apparent by this figure and/or its legend. Cutting it off at 0.4 doesn't tell me very much about what is going on here. I don't think that "NOx" was used previously in the paper, so shoud be defined in the figure legend. Nutrients are incorrectly labeled again (missing charges etc). I would suggest just writing out the names if it is difficult to properly add superscripts etc. in the software used for the figures.

Minor comments: Line 151: change "was" to "were" Line 199: change "sunlight" to "solar" Line 323: "MWH-UniP1" Add "related" to the end of this OTU designation. Line 329: Use the correct designation for phosphate Line 349: Add (BR) after bacterial respiration to define the acronym. Line 375: Citations needed for "some studies". Line 442: change "caused" to "was related to"

---

## Author Comment (AC1) · 7 Jul 2016

**Responses to Anonymous Referee #2**

**R#2:** The study by Vaquer-Sunyer describes the effects of wastewater treatment plants effluent inputs on different microbial parameters. I find the topic interesting and relevant for publication in BG. However, the approaches used are not well explained and therefore it is hard to judge the results. Moreover, I miss precise definitions and statements in the discussion. Therefore the manuscript seems in the current status premature.

**Comment (C):** The manuscript has changed considerably since its first version thanks to the comments by 3 reviewers, including yours. We believe that the manuscript has improved significantly and hope you will find it suitable for publication in its present form. Please, see detailed comments below.

Introduction
**R#2:** Line 50 ff. The introduction (and in the discussion) gives a very anthropogenic view about hypoxia. Various prokaryotic groups' life in oxygen minimum zones and often exists only at those conditions (read for example "Microbial ecology of expanding oxygen minimum zones" Jody J. Wright, Kishori M. Konwar & Steven J. Hallam). From a microbial point of view the biological diversity therefore increases by the presence of oxygen minimum zones. Moreover, oxygen minimum zones in the central Baltic Sea are connected to the presence of a halocline that prevents water mixing in spring and autumn and therefore a typical phenomenon in the Baltic Sea. The authors mean coastal water hypoxia (as in the cited Conley 2011 publication) that should become clearer in the introduction (and discussion).

**Comment (C):** We meant eutrophication-driven hypoxia, typical in coastal waters.

**Action (A):** We include some sentences to make clear that Baltic Sea suffers from eutrophication-driven hypoxia. We also acknowledge that in oxygen minimum zones prokaryotic diversity could increase and refer to the above-mentioned paper. The text now reads (lines 51-52 and 55-59): "The Baltic Sea has the largest area affected by **eutrophication-driven hypoxia** (Conley et al., 2011). (…) The lack of oxygen in marine waters causes death of the marine organisms and catastrophic changes in marine **metazoan** communities. Thus, hypoxia is emerging as a major threat to marine biodiversity (Vaquer-Sunyer and Duarte, 2008), although prokaryotic diversity can increase in oxygen minimum zones (Wright et al., 2012)."

Methods:
**R#2:** Line 105-105: what salt solution was used?

**A:** We now refer to the paper used to make the salt solution (Søndergaard et al., 2003).

**R#2:** Line 131-133. Use space between unit and number.

**A:** We have done so

**R#2:** Line 136: please give a short description who the calculation of the metabolic rates were performed. If I understand correctly, CR was estimated from change in oxygen over night and GPP from the sum of NCP and CR. Since DOC is produced during daytime the CR can differ between night and day. Typically incubations are performed at light and dark conditions in parallel to estimate CR and GPP. Please discuss that your approach gives comparable results as the parallel incubations.

**C:** We agree with the reviewer that a description on how metabolic rates were calculated will improve the manuscript. Dark incubations can underestimate CR for two reasons: (i) respiration during daylight is probably higher than at night and (ii) under dark conditions phytoplankton growth is suppressed, so contribution of phytoplankton to community respiration is limited. Here, we assume equal respiration rates during day and night, as it is done by incubations under light and dark conditions. However, as communities are incubated under conditions mimicking natural conditions, phytoplankton respiration will contribute more to community respiration than under light/dark incubations.

**A:** We have included a short description on how the metabolic rates were calculated. The text

now reads (lines 140-144): "NCP was estimated as the changes in dissolved oxygen content during 24 hours intervals (dDO/dt). CR was calculated from the rate of change in DO during the night from half an hour after lights went of to half an hour before light went on. CR was assumed to be the same during light and dark. NCP in darkness equals CR during night. GPP was estimated as the sum of NCP and CR (GPP = NCP + CR)." We have also included discussion about the possible differences in CR during day and night and comparability with light/dark incubations (lines 148-156): "As incubations were performed following a natural light regime to mimic natural conditions, results may differ from incubations performed at light and dark conditions in parallel. Both approaches assume equal respiration rates under light and dark conditions. This assumption may lead to underestimate CR and GPP, as respiration rates are probably higher during daylight than at night (Grande et al., 1989; Pace and Prairie, 2005; Pringault et al., 2007), but it does not affect NCP estimates (Cole et al., 2000). In incubations performed under dark conditions, phytoplankton growth is suppressed, decreasing phytoplankton respiration contribution to community respiration."

**R#2:** Line 151: The 341-805r primers were designed for bacteria the protocols in Hugerth et al are for eukaryotes. If I understand correctly a two step PCR were performed, please describe the protocol in more detail, especially if the PCR contained several independent PCR cycles. This is important information since this can introduce a strong bias into the abundances estimates for bacteria.

**C:** The 341f-805r primers are designed for bacteria and were first used for bacteria in: Herlemann, D. P., et al. (2011). "Transitions in bacterial communities along the 2000 km salinity gradient of the Baltic Sea." ISME J. However, that study was made with 454-pyrosequencing. The description in Hugerth et al. is for Illumina Miseq designed primers using standard Nextera primers as template. The PCR program applied has two steps to limit biases such as described previously for pyrosequencing: Berry, D., et al. (2011). "Barcoded Primers Used in Multiplex Amplicon Pyrosequencing Bias Amplification." Applied and Environmental Microbiology 77(21): 7846-7849, and recently used in e.g.: Savio, D., Sinclair, L., Ijaz, U. Z., Parajka, J., Reischer, G. H., Stadler, P., Blaschke, A. P., Blöschl, G., Mach, R. L., Kirschner, A. K. T., Farnleitner, A. H. and Eiler, A. (2015), Bacterial diversity along a 2600 km river continuum. Environmental Microbiology, 17: 4994–5007. In the latter paper the two-step PCR was applied for the Illumina Miseq platform. Thus, in our work, the first step in the two-step PCR uses the main primer set 341f-805r to amplify the correct fragment. The second step is applied to attach the handles and indexes needed to run the 16S Illumina Miseq run and for tagging/barcoding individual samples.

**A:** We have now clarified the PCR method in Material and Methods section as follows (lines 201-205: "…with some modifications. We thus performed a two-step PCR: (i) amplification with the main forward and reverse primers 341F-805R to amplify the correct fragment within the V3-V4 hypervariable region of the 16S rRNA gene; (ii) amplification using template from the first PCR to attach the handles and indexes needed to run the Illumina Miseq run and for barcoding individual samples."

**R#2:** Line 119: What happened to the data of the DPA measurements?

**A:** DPA concentration for effluent and seawater are given in tables 1 and 2, and for each experiment are included in table S1. As DPA did not explain variation in metabolic rates or bacterial community composition it is not discussed in the results or discussion sections.

Result
**R#2:** Line 190 221 table 1 and 2: How was DOC and TOC measured (replicates, sensitivity,. . .)?
     **C:** We only measured DOC as all samples have been filtered. Samples were taken in duplicate.
     **A:** We have better explained how DOC was measured. The text now reads (lines 172-179 and 182-185): "Samples for chlorophyll a (*Chl.a*), dissolved organic carbon (DOC) and nutrients were taken on days 0, 1, 3, 5 and 7 from the two 2.3 L bottles for each treatment incubated simultaneously than the bottles used to monitor oxygen content changes to calculate metabolic rates. Samples were taken in duplicate. For the last day of the experiment (day 7) the 2 bottles used to monitor oxygen content were used to sample *Chl.a* and nutrient content. Samples for nutrient determination were filtered using pre-combusted (450ºC, 4 h) glass-fiber (GF/F Whatman) filters and 0.2 μm membrane filters and frozen until analysis. All equipment used for handling the samples was acid washed." "DOC was measured on a Shimadzu TOC V-CPN in non-purgeable organic carbon (NPOC) mode on

acidified samples (HCl to pH <2). The instrument was calibrated daily with potassium hydrogen phthalate. DOC concentrations were calculated from the average area of 3 injections, with an area covariance of less than 2%."

**R#2:** Line 273: How was the change in the number of OTUs?

     **A:** For comparison we have now added text and a new supplementary Figure 2 that show variation in observed number of OTUs (Fig. S2A) and Chao.1 richness index (Fig. S2B). The text now reads as follows (Line 341-345): "Moreover, we analyzed the richness and found that the observed number of OTUs ranged between 206-946 ± 171 and Chao.1 index values ranged between 306-1273±220. Richness was generally lower in effluent amended treatments compared to controls, except for in the April experiment."

Discussion
**R#2:** Line 345: It is unclear what the abbreviation DOM stands for. Typically it is used for dissolved organic matter (which was not measured in this study), but in the manuscript it reads rather that N-rich dissolved organic matter is meant. Please clarify. The results suggest that certain concentrations of nutrients (Figure 2; figure 4, Figure 5 1:5; IN) cause a change in the GPP, CRR and BP compared to the control treatment. Moreover the statistics give often DOC and CHla as the variances explaining the variability (line 224, line 234, line2 44) and only for bacterial production a correlation with NO3. Therefore the statement: " OM significantly increased bacterial production, whereas it decreased gross and net primary production and community respiration rates" seems not justified.

     **C:** We used DOC as a proxy for DOM. Mixed effects models showed that DOC was significantly related to BP, GPP, NCP and CR, with high $R^2$ values.

     **A:** We have included a sentence to explain that we used DOC as proxy for DOM. The text now reads: (Lines 222-230): "Metabolic rates data from the four experiments were combined to test the relationship between the given metabolic rates and physicochemical parameters (Table 1) by mixed effects models. Physicochemical parameters were selected avoiding collinearity. Selected variables were DOC, DON, nitrate and phosphate concentration. We used DOC as a proxy for dissolved organic matter (DOM). Variables were selected according to its significance. Variables were removed following its p value (i.e. variables with higher p value were removed first) until all parameters were significant). To account for pseudo-replication we used incubation day nested to season (i.e. experiment) as a random factor." (…) Lines 415-418: "DOM significantly increased bacterial production, whereas it decreased gross and net primary production and community respiration rates, as showed in the results of the mixed effects models where DOC is used as a proxy for DOM."

**R#2:** Line 413. Alpha diversity is not only expressed in the Shannon index. I think you mean Shannon index here, but it would be interesting to see how the total number of OTUs (richness) change.

     **A:** We have now changed the text to also accommodate our new analysis of species richness (observed number of OTUs and Chao.1 index). Lines 538-542: "The observed effect of species loss, i.e. lower richness (observed number of OTUs and Chao.1 index) and Shannon diversity index, may be closely linked with the functioning of microbial communities and could potentially render the whole community more sensitive to environmental perturbations (Allison and Martiny, 2008; Bell et al., 2005; Loreau, 2000, 2004; Shade et al., 2012)."

**R#2:** Please discuss also the artifacts that can arise from long term bottle incubations especially for t7.

     **A:** We have added a paragraph on the "bottle effect" to address this issue (lines 549-562). For further detail, see reply to comment by reviewer 1 on bottle effect.

**R#2:** P has also a strong impact on the eutrophication in the Baltic Sea since many Cyanobacteria are able to fix N (read for example "Andersson, A., Höglander, H., Karlsson, C., and Huseby, S. (2015). Key role of phosphorus and nitrogen in regulating cyanobacterial community composition in the northern Baltic Sea. Estuar. Coast. Shelf Sci. 164, 161–171."). Since P was also measured in the experiments, and found to have a strong influence on the BCC (Line 271), I wonder why P is rarely

discussed.

**A:** We now include a sentence in the discussion section including reference to the above-mentioned paper. The text now reads (lines 507-5349: ""Apart from the influence of temperature in structuring the bacterial communities in the present paper, shifts in bacterioplankton community composition was highly correlated with changes in phosphate concentrations. In agreement, previous findings show that phosphate is a driver of shifts in community structure in the Southern Californian coast and Baltic Sea (Fuhrman et al. 2006; Andersson et al. 2010). For example, Andersson and colleagues (2010) suggested that limiting conditions due to a decline in phosphate during the summer Cyanobacterial bloom promote selection in the bacterioplankton community where specific OTUs can proliferate. Moreover, in an adjacent area of the Baltic Sea Proper opportunistic cyanobacteria, including $N_2$-fixers and picocyanobacteria, proliferated despite low phosphorus concentrations and may instead have been fueled by bioavailable nutrients from filamentous Cyanobacteria (Bertos-Fortis 2016). Recent evidence suggests that availability of phosphorus has a substantial impact on eutrophication in the Baltic Sea since many Cyanobacteria are able to fix nitrogen (Andersson et al. 2015). In the present study phosphate concentrations showed small variations between treatments within each experiment and we observed primarily seasonal oscillations between experiments. Absolute shifts in composition among the groups Bacteroidetes, Betaproteobacteria and Alphaproteobacteria were positively correlated with absolute changes in phosphate whereas shifts in Planctomycetes, Verrucomicrobia and Cyanobacteria were negatively correlated with variation in phosphate. Nevertheless, changes in phosphate concentrations significantly explained variation in community structure within the July experiment. Hence, the communities responded to effluent inputs by shifts in species composition and the influence of seasonal changes in phosphorus concentrations was outweighed by the simulated environmental disturbance investigated here. Thus, long-term changes in phosphorus resulting from natural seasonal variation or climate change related effects accompanied by episodic short-term effluent inputs may form a synergistic permanent impact on the structure of bacterioplankton communities with severe consequences for ecosystem services."

Figures
The legend of figure 8 is almost not readable. The colors in the plot are not easy to distinguish the abbreviations should also be given in the legend

**A:** We have now modified the figure to make it more clear and we have increased font sizes.

**R#2:** Figure 2 and Figure 4: they are based on the same data, whereas one contains subtracted CR? Therefore I think one of them is sufficient.
**C:** Although the reviewer is right regarding that GPP and NCP are derived from same data and NCP equals GPP – CR (with CR being positive), we do believe that including both figures is important to understand metabolic rates dynamics. Net community production gives information on whether planktonic communities are heterotrophic (if NCP is < 0) or autotrophic (when NCP > 0) o if there is metabolic balance.

---

## Author Comment (AC3) · 7 Jul 2016

**Responses to Anonymous Referee #1**

**Comments**

**R#1:**The authors tested the effect of the wastewater treatment plant (WWTP) effluent inputs on Baltic Sea planktonic communities in 4 experiments. They did so sampling seawater during winter, spring, summer and autumn and observing the effect of different WWTP addition to natural communities. They observed that nitrogen-rich DOM inputs increased BP and decreased PP. This trend will drive to an increase of carbon consumption and shift the ecosystem toward heterotrophy. Although the experiment was well performed and that the authors analysed several variables, I had the feeling that the paper was written in a hurry, sometimes with lack of precision and details. However, the study and the results obtained here are strongly recommended to be available for scientific community. Then, I recommend this manuscript to be published with minor revisions.

**Comment (C):** We have made extensive changes following recommendations by 3 reviewers. We believe that the manuscript has improved considerably after incorporating all changes suggested by the reviewers. We hope that you find it suitable for publication.

**Detailed comments:**

**Introduction**

**R#1:** 65: For non-specialists in WWTP, I think that you should explain why you are talking about the Chesapeake Bay and its discharge limit. Is it a bay that receives important WWTP effluent at the US scale? Comparable with the study area?

**C:** We included Chesapeake Bay discharge limits because it's a bay with serious problems of lack of oxygen, like the Baltic Sea. It's an enclosed bay, and the Baltic Sea is an enclosed sea. We tried to highlight that in sensitive areas discharge limits could be stricter.

Action (A): We have included an explanation about the inclusion of discharge limits in Chesapeake Bay. The text reads (lines 67-70): "(...) Chesapeake Bay, the largest U.S. estuary that experiences severe hypoxic conditions, discharge limits (...) Both areas, the Baltic Sea and Chesapeake Bay, are enclosed water bodies with excessive anthropogenic nutrient inputs."

**R#1:** 58-73: You should explain a little bit more how is the WWTP effluent in the Baltic Sea. What is its average discharge? TN, DIM, DON, DOM concentrations? Do any treatment have been implemented in order to reduce the WWTP discharges into the Baltic Sea?

A: We have now further explained WWTP effluents characteristics. The text now reads (lines 70-78): "Wastewater treatment plants contribute 10-20% of total nutrient loading in the Baltic Sea (Hautakangas et al., 2014). Estimates of total nitrogen loads to the Baltic Sea due to WWTP effluents are about 110 000 tons of nitrogen per year, and for total phosphorus loads are around 11 000 tones of phosphorus per year (Hautakangas et al., 2014). Some Baltic countries have implemented nutrient reductions in their WWTP. Denmark and Germany have reduced both nitrogen and phosphorus loadings significantly. Sweden and Finland have reduced phosphorus loads but have failed so far in reducing nitrogen loads down to 70% as recommended by HELCOM (2009) (Hautakangas et al., 2014). "

**R#1:** Methods 87-90: Why did you choose to sample during the four seasons? Does the effluent discharge more during winter due to rainfall and enhance the WWTP inputs on Baltic Sea?

**C:** Environmental conditions differ between seasons (i.e. nutrient or DOC concentration). The amount of TN and DON differs between seasons, as it can be seen in table 2. Also, planktonic community differs between seasons too, and it can influence its responses to WWTP inputs. We also sampled during the four seasons to be able to acquire the full year variability.

A: We have explained in the text why we sampled during the 4 seasons. The text reads (lines 111-112): "(...) to be able to measure seasonal variation in both planktonic communities and effluent characteristics under different environmental conditions."

**R#1:** 93-96: Did you check your WWTP after being filtered and frozen if some organism remains into your sample (as some bacteria may be smaller than  $0.2 \mu m$ ) using for example flow cytometry?

C: We measured bacterial production in the WWTP water source for the experiment conducted in spring to test if some bacteria remained in the WWTP water. BP values were very low, lower than BP

in autoclaved milli-Q water for the same day (DMP 125.25 for WWTP and 200.53 for autoclaved milli-Q).

**R#1:** 102-103: In my opinion, the third treatment is unclear. 103-105: I am not sure that I understood how did you do CD. You said that CD consisting of seawater diluted with milli-Q to have the same portion of community that 1:10, 1:15 and IN. Does that mean that you made the same "dilution" than the WWTP treatment but instead of using WWTP, you used milliQ and you made a CD of 1:10 (100mL of milliq for 1000ml of seawater), another of 1:5 (200ml of milliq for 1000ml of seawater) and another as IN (?)? If this is the way you made, I don't understand how do you have just one CD...

**C:** We agree with the reviewer that the explanation of how the treatments were made was unclear in the previous version of the manuscript.

A: We have better explained the preparation of the different treatments. The text now reads (lines 116-119): "Those 3 treatments (1:10, 1:5 and IN) were performed to contain the same portion of community, so the 1:10 and the IN treatments were diluted with autoclaved milli-Q and salt solution to obtain the same community portion than the 1:5 treatment."

**R#1:** 122-137: In this paragraph, you didn't explain how did you do dark and light incubation, how many replicates did you have. You just said "bottles were incubated at the in situ temperature... during one week". Confinement methods prone to error as you might exclude zooplankton, enhance trophic interactions within the bottle and so on. Most of the incubations realized to estimate changes in DO in order to determine NCP and CR lasts between 12-48h. One week of incubation is a lot. Do you think that the community inside your bottles after 3-4 days of incubation is representative of in situ community that may receive once an amount of DON or DIN 3-4 days earlier? You did not talk in this paragraph about the confinement effect and the effect of 7-days incubation. I think that it is really important and that you should take time to write about it and give some criticism of your estimates.

**C:** Bottles were incubated under light/dark conditions in 2.3 L duplicate transparent bottles. We performed a pilot experiment lasting 20 days to decide how long the incubations should last. One week is a commonly used time for experimental incubations in micro/mescosms. In addition, water temperature in the Baltic is low, and a long period of time was needed to see changes in the metabolic rates and species composition due to relatively slow growth rates. Our microbial communities showed no growth of opportunistic OTUs, showing that the community present in our experiments was representative of in situ community.

A: We have included a better explanation in the number of replicates. The text now reads (lines 127-133): "Water from the respective treatments was siphoned carefully to avoid bubble formation into four 2.3 L glass bottles per treatment sealed with gas tight stoppers. Bottles were incubated at the in situ temperature (Tables 1 and S1) in a temperature-controlled chamber during one week. Oxygen was measured every minute in 2 of the 4 replicate bottles using optical oxygen sensors (optodes) and a 10channel fiber optic oxygen transmitter (oxy-10, PreSens®). The remaining 2 bottles per treatment were used to sample nutrient and chlorophyll a concentrations." We have also added a paragraph to address the potential effects of incubation. The paragraph is as follows (Line 549-562): "The so-called "bottleeffect", in which confinement of water causes shifts in bacterioplankton community composition and physiological rates, is a factor to consider in interpreting results from experiments with natural microbial assemblages (Fuchs et al., 2000; Massana et al., 2001; Baltar et al., 2012). Such effects are typically detected by rapidly increasing proportions of fast-growing gammaproteobacterial populations and rate measurements across all treatments (including controls) (Pinhassi and Berman, 2003; Sjöstedt et al., 2012; Dinasquet et al., 2013). In our current experiments, microbial community composition remained relatively similar to in situ communities and we did not observe excessive increases in opportunistic bacterial populations in the controls. Rather, increases and decreases in relative abundance were observed among populations typical of Baltic Sea Proper, such as Rhodobacteraceae, Synechococcus and BAL58 (Lindh et al., 2015). Thus, although confinement per se surely had effects on microbial diversity and rates, our results indicate that such effects were minor relative to the actual treatment effects."

**R#1:** 130-133: You explained that incubations were illuminated by artificial light with a mean PAR of 1373.2 uW/cm2. Why did you choose this amount of light? Is it representative of the daily PAR that Baltic community receives at 2m depth? Does the illumination was constant during light exposition or does it increase until reaching a maximum natural irradiance and then decrease? The light hours range

for the summer experiment is about the double of light for winter experiment. Do you think that GPP is comparable in summer and in winter experiment? Would it not be better to express GPP per hour and not per day?.

**C:** Irradiation intensity was a constant 1373 uW/cm2 during the daylight hours of the experiment, as an artificial ramping up and down of light was not possible in our incubation room. Daylight hours were determined to match the daylight hours of the season in Sweden, since the goal was to understand the system during the seasonal cycle. Natural irradiation at high latitudes, such as in Sweden, has large variations in both intensity and duration (i.e. daylight hours). The irradiation rate here is equivalent to that received at 2.5m depth in the winter and 7m depth in the summer, at Kalmar, Sweden, the location at which the samples were collected.

As summer has much longer daylight hours, it has higher GPP in spring and summer than during winter and autumn. GPP is usually reported per day and not per hour. We believe that reporting it per day helps to see the seasonal differences of this metabolic rate.

Incident irradiation values in Kalmar, Sweden were obtained from the Strång model, from the Swedish Meterological and Hydrological Institute (SMHI).

http://strang.smhi.se/extraction/index.php?data=tmsrs&lev=2

A: We have added the depth information to the manuscript, and deleted the word "mean" describing the PAR dose, as it may have been misleading to the reader, since the dose was constant (lines 137-139): "This irradiation dose corresponds to the irradiation received at a depth of 2.5 m in the winter and 7 m in the summer, at Kalmar, Sweden."

**R#1:** 134-137: I understood that you estimated NCP this way: DOmin1-DOmin0, DOmin2-DOmin1, DOmin3-DOmin2, . . . DOmin1440-DOmin1439. Then, you had NCPmin1, NCPmin2, NCPmin3, . . ., NCPmin1440 and you sum it to have it per day. Is it right? Or did you make it directly DOmin1440-DOmin0=NCP24h? According to the calculation that you made, did you compare it with the other way? Is it similar?

**C:** We estimated NCP as dDO/dt. Just using DOmin1440-DOmin0 would lead to loss detailed information that we had, as we measure dissolved oxygen every minute.

A: We have added text to better explain how metabolic rates were calculated. The text now reads (lines 140-144): "NCP was estimated as the changes in dissolved oxygen content during 24 hours intervals (dDO/dt). CR was calculated from the rate of change in DO during the night from half an hour after lights went of to half an hour before light went on. CR was assumed to be the same during light and dark. NCP in darkness equals CR during night. GPP was estimated as the sum of NCP and CR (GPP = NCP + CR)."

**R#1:** 141: Can you explain what is the killed control please?

A: We now explain how the killed control was made. The text now reads (lines 160): "1 killed control with 5% trichloroacetic acid (TCA)"

**R#1:** 140-142: It is not specified during the 60min incubation the temperature and irradiance received by the samples.

A: We now explain that BP samples were incubated in the same conditions than the rest of the samples. The text reads (lines 161-163): "(...) in the temperature-controlled room, at the same incubation temperature and light irradiance than the rest of the samples."

**Results**

**R#1:** 211-245: In the Metabolic rates results section, I missed to read more about statistics. For example, does GPP was significantly different between each treatment for each season? The same for CR and NCP.

C: Our statistical approach consists of using mixed effects models to test which

physicochemical parameters influenced metabolic rates. We cannot use ANOVA tests to test for differences between treatments for each experiment as there was huge temporal variability and the number of replicates was only 2. Due to the limited replicates we cannot use repeated measures MANOVA for those metabolic rates. We could use repeated measures MANOVA for BP as there were 3 replicates for each treatment and day.

**R#1:** 223: I think you wanted to write "GPP decreased with Chl a content, it increased with DOC concentration" in reference with Table 3.

**C:** There was a mistake in the previous version of the manuscript, where Chl.a and DOC were changed.

A: We have corrected the typo. We have done mixed effects again to include sampling day nested to season as random factor. Now, only DOC concentration is significant in the mixed effect model. We have removed chlorophyll a from the table.

**Discussion**

**R#1:** 345-348: I don't agree with this part of your discussion. I really don't see from which results you concluded that DOM significantly increased BP and decreased NCP, GPP and CR. Where are the statistical tests showing that GPP, NCP and CR were significantly different from control and CD? I understand that in BP results (247-260), you observed a tendency to increase with higher addition of effluent (nothing significant I guess), that you observed significant differences in BP for different sampling day, treatments, interaction between sampling day and treatment but there is no mention in this paragraph that BP was significantly higher at the end of the experiment with effluent addition... Furthermore you observed no significant differences between treatments for spring and winter experiments. I don't see in this results section any mention of "BP was significantly higher under effluent addition than control and CD at the end of the experiment" For GPP, NCP and CR, you even didn't show any statistical test in the results that can lead you to this conclusion. In Figs. 2, 4 and 5, we can see that the responses are different according to the season, that responses aren't linear along the experiment (response different at day 3 than at day 6) but you didn't talk about that in the result section and we missed that. Looking to these figures, I don't agree with the general conclusion that NCP, GPP and CR were suppressed by DOM addition, but if some statistical tests show me the inverse ok... but I need to see it! And, did you consider that the "good" response is at day 7? If it is so, why? Why not at day 3?

**C:** We conclude that DOM significantly increased BP and decreased NCP, GPP and CR from the results of the mixed effects models. We used DOC as a proxy for DOM. Mixed effects models showed that BP increased with the concentration of DOC ( $R^2$ = 0.91, p < 0.0001), and that NCP, CR and GPP decreased with DOC content ( $R^2$ = 0.79, p < 0.0001,  $R^2$ = 0.84, p < 0.0001 and  $R^2$ = 0.84, p < 0.0001, respectively). The estimate of the mixed effects models represents the slope associated to the given variable; p values were calculated comparing nested models with and without the inclusion of the response variable (i.e. ANOVA for the different models including or not the response variable). DOC concentration significantly decreased GPP and CR by itself without taking into consideration the other variables used in the models (i.e. random factor). DOC content significantly increased BP in summer, spring and winter, but not in autumn (see graphs bellow). NCP decreased with increasing DOC content, but this relationship was not significant when omitting the random factor included in the mixed effects model. Bellow you can find figures showing the relationships and the R2 and p values for those relationships.

---

## Author Comment (AC4) · 7 Jul 2016

[revised manuscript text omitted]

Figure 1

[Figure]

Figure 2

[Figure]

Figure 3

[Figure]

Figure 4

[Figure]

Figure 5

[Figure]

Figure 6

[Figure]

Figure 7

[Figure]

Figure 8

[Figure]

Figure 9

---

## Author Comment (AC5) · 7 Jul 2016

Supplementary Material for

[revised manuscript text omitted]

---

## Author Comment (AC2)

**Responses to Anonymous Referee #3**

General Comments:
The current study by Vaquer-Sunyer and colleagues describes the effects of wastewater treatment plant effluent inputs to the Baltic Sea on coastal planktonic microbial communities. The topic should be of interest to readers of Biogeosciences, and should be able to be made appropriate for publication after revision. The authors tested a number of relevant microbial parameters, and the experiments appear to be well-thoughtout and executed, although some of the methods require some clarification. The main results showed an increase in bacterial production and decreases in primary production and community respiration following amendment with wastewater, along with some changes in bacterial community composition. There is some confusion, especially during the discussion section, between discussion of metabolic activity vs. community composition - i.e. it seems that an increase in BP and decrease in PP is taken to indicate a shift in community type (autotrophic to heterotrophic), which was not tested or substantiated by the data in the paper. I find that the discussion section in particular becomes somewhat disjointed, and that some of the conclusions drawn from the study are overstated (i.e. presented with more certainty than the data allow). As such, the paper requires more thought and more careful presentation before it is ready for publication. I hope that specific comments below are helpful in this regard.

   **Comment (C):** We have carefully considered the constructive comments by the 3 reviewers in preparing the revised version of the manuscript and have made, accordingly, extensive changes. We have revised the original version to address all of the comments raised by the reviewers. We believe, that as a result of these changes, the manuscript is now much improved relative to that originally one submitted, and hope that you will find it now acceptable for publication in Biogeosciences.

Specific Comments:
**R#3:** Introduction: Line 67-69: This statement regarding reduction of TN seems quite specific. Can a reference be provided?
   **Action (A):** We have referred to Bronk et al. (2010).

**R#3:** Methods: Line 127: "in situ temperature" - are these the temperatures listed in Table 1? If so, please refer to Table 1 here.
   **A:** We now refer to table 1 and supplementary table S1 were measured incubation temperatures are reported.

**R#3:** Line 136: A description of the method to calculate metabolic rates (even if it is an abbreviated summary) should be provided here, not simply a reference to another paper that describes the method. I looked up the other paper, and it is not clear to me how all of the metabolic rates were derived from the data in the current paper.
   **C:** We agree with the reviewer that including a brief description of the method used to calculate metabolic rates would improve the manuscript.
   **A:** We included a brief description of the method used to calculate metabolic rates. The text now reads (lines 140-144): "NCP was estimated as the changes in dissolved oxygen content during 24 hours intervals (dDO/dt). CR was calculated from the rate of change in DO during the night from half an hour after lights went of to half an hour before light went on. CR was assumed to be the same during light and dark. NCP in darkness equals CR during night. GPP was estimated as the sum of NCP and CR (GPP = NCP + CR)."

**R#3:** Line 136: "water properties" - please either list all of the properties (maybe a better term would "physicochemical parameters") used in the statistical models, or refer to the table that contains them.
   **A:** We changed the term by "physicochemical parameters" and refer to table 1 as suggested by the reviewer.

**R#3:** Line 154: What are the "biological replicates"? I did not get this from the description of the experimental/treatment design. Given this, I think that the treatment description (Line 97 onwards) needs to be improved - I see four experiments (one for each season) with five treatments each, but no replicates. Perhaps a list of everything that was tested for each treatment within each experiment should be included. It's not clear to me what exactly was measured on which sample.
   **C:** We agree with the reviewer that the description of the replicates and how samples were

taken needs to be improved.

A: We have included all information on the number of replicates and sampling for different parameters. The text now reads (lines 127-133): "Water from the respective treatments was siphoned carefully to avoid bubble formation into four 2.3 L glass bottles per treatment sealed with gas tight stoppers. Bottles were incubated at the in situ temperature (Tables 1 and S1) in a temperature-controlled chamber during one week. Oxygen was measured every minute in 2 of the 4 replicate bottles using optical oxygen sensors (optodes) and a 10-channel fiber optic oxygen transmitter (oxy-10, PreSens®). The remaining 2 bottles per treatment were used to sample nutrient and chlorophyll a concentration." Lines 172-179: "Samples for chlorophyll a (*Chl.a*), dissolved organic carbon (DOC) and nutrients were taken on days 0, 1, 3, 5 and 7 from the two 2.3 L bottles for each treatment incubated simultaneously than the bottles used to monitor oxygen content changes to calculate metabolic rates. Samples were taken in duplicate. For the last day of the experiment (day 7) the 2 bottles used to monitor oxygen content were used to sample *Chl.a*, DOC and nutrient content. Samples for nutrient determination were filtered using pre-combusted (450ºC, 4 h) glass-fiber (GF/F Whatman) filters and 0.2 µm membrane filters and frozen until analysis. All equipment used for handling the samples was acid washed."

Results:
R#3: Line 186: Can you please clarify whether the nutrient determinations were done on the samples collected for each expieriment, following the filtration and freezing steps described in the Methods? I suggest making this clarification in the Methods so that the reader knows exactly where the reported data are coming from.

C: We have clarified how samples for nutrients, chlorophyll a and DOC were collected and handle. Please, see above.

R#3: Line 192: The Methods section should be updated to include how the seawater samples for nutrient and chlorophyll analysis were collected and handled. I see a description of planktonic microbial community sample collection only. The description of how the samples were analyzed for nutrient and chl content, is complete, just not collection, filtration, storage, etc.

C: Please, see above.

R#3: Line 203: "as a consequence of re-mineralization" is probably a good assumption to describe increasing nutrients, but because the source of the increased nutrients was not tested in the current study, this statement (and its degree of certainty) is not appropriate for the Results section.

A: We have deleted "as a consequence of re-mineralization" as suggested by the reviewer.

R#3: Lines 204-209: I do not see a description in the "statistical methods" section of the Methods that could have been used to arrive at these conclusions regarding cal. The methods seem to cover metabolic rates and community structure, but no the relationships among physicochemical variables such as Chl and light. Please clarify this in the Methods.

A: We have now included in the "statistical methods" all the approaches we have used in the manuscript. The text now reads (lines 219-221): "Relationships between chlorophyll a content and physicochemical parameters (nitrate concentration, light hours and temperature) were tested by fitting ordinary least square regression."

R#3: Line 255: Rather than saying that BP 'depended on' DOC, it might be more useful to describe the direction of the relationship.

A: We have done so. The text now reads (line 320): "BP was positively correlated to DOC content (…)"

R#3: Line 267: "temperature significantly explained..." I question whether strong conclusions can be drawn regarding the influence of temperature. Given the range of temperatures (3,4,7, 18), it seems that the single high temperature (18) is an outlier and would exert extra influence on the correlations in the Mantel test. Can you address this in this review response, since many of the relationships in the paper seem to revolve around temperature?

C: The reviewer is correct in that outliers affect the outcome of any MANTEL test. However, in the current paper, temperature ranged from 3-18°C because the experiments were performed at different seasons. We performed the MANTEL tests on all samples from all experiments and analyzed the bacterial communities collectively. Even though the large span up to 18°C in summer lead to large shifts in community composition we do not consider this high temperature as outliers. Instead, we think

that the results are expected, as temperature is a major driver of shifts in community composition. From this we instead conclude that despite large shifts in community composition due to changes in temperature for all communities, there are still distinct differences between controls and effluent inputs, especially in the July experiment.

**R#3:** Line 273: "relatively similar" is unclear. Perhaps provide the range in alpha diversity across all experiments in parentheses and say "similar".

    **A:** To clarify the sentence we have now modified the text so that it reads (Lines 338-339): "Alpha diversity estimated from Shannon diversity index was relatively similar between treatments in each experiment and ranged from 3.34 - 5.82 ± 0.51 (fig. 7)."

**R#3:** Line 275-276: Can you clarify the wording please? I think what is meant is "a lower Shannon index was observed for all nutrient treatments compared to the controls", but I am not certain based on current wording.

    **A:** We have now changed this sentence to (Lines 340-341): "Nevertheless, a lower Shannon index was observed for all nutrient treatments compared to the controls in all experiments except April (fig. 7)."

**R#3:** Line 281-282: Is the implication here that the Betaproteobacteria decreased in the control over time, rather than increasing in the treatments? That is interesting, and I suggest making clear what the conclusion related to this result is. Also, if there were changes in the control during the experiment, is there concern over bottle effects?

    **C:** Betaproteobacteria decreased in relative abundance in the control compared to the other treatments over time.

    **A:** We have now rephrased the text (Lines 349-351) to make it more clear: "Nevertheless, Betaproteobacteria decreased in relative abundance by more than half in controls until T7 while they maintained their abundance in the other treatments". We have now also added a new paragraph to address the potential issue of bottle effects (Lines 549-562). See comment for Reviewer 1.

**R#3:** Lines 284-285: "higher relative abundance" - Can you please add in parethenses how much higher the relative abundance was, compared to other treatments and controls (on average)? Also, is there statistical significance associated with this statement? It is fine if there is not, but I still suggest providing some numbers so that the reader can make the comparison more clearly.

    **A:** We have now added numbers to show the reader differences in relative abundance of the group others that dominated 1:5 treatments. The text now reads as follows (Lines 352-354): "Bacterial groups other than the 8 major phyla/class ("Others") had nearly four-fold higher relative abundance in the 1:5 treatment compared to the other treatments and the controls."

**R#3:** Line 290: "increased in the control" Same question as above - With so many changes in the control, are we just seeing bottle effects over time? Can you comment on the validity of comparing these long incubations? Why would things be changing in the control?

    **C:** There are often changes in controls for micro/mesocosm experiments simply because of temporal succession and/or that the available nutrients are being utilized in the incubations. Therefore, in any given experiment the comparisons are being made between the observed successions in the treatments compared to controls.

    **A:** To clarify the importance of controls we have added a paragraph (Lines 549-562) on the "bottle-effect" (see comment to reviewer 1).

**R#3:** Line 301: Can you please define what is meant by "finer phylogenetic scales"? i.e. at OTU level? Phylum level?

    **A:** We have now modified the text to clarify this definition and the sentence reads (Lines 368-370): "Hence, we performed Pearson correlation tests to determine links between environmental factors, metabolic rates and shifts in relative abundances at phyla/class level".

**R#3:** Line 301: "when communities responded to experimental treatments" I'm not sure what this means. Can you clarify whether you mean that you only looked at links between environmental and biological factors in experiments where there was a response to the treatment? Perhaps this needs to be split into more than one sentence to make the meaning clearer.

    **A:** We have now changed the text, see previous comment.

**R#3:** Line 302: "were positively correlated" Is this referring to the relative abundances of these groups? Can you please say what about these groups was correlated with temperature?

**A:** The text now reads (Lines 370-372): "Shifts in relative abundances of Cyanobacteria, Planctomycetes and Verrucomicrobia were positively correlated with changes in temperature (fig. 9)".

**R#3:** Line 309: It is not clear to me where the explanation of the variance is coming from here. Earlier in the paragraph, Pearson correlation is referred to, but I am not sure that makes sense here. Can you please specify?

**A:** We have now modified the text to make it more clear (Lines 377-378): "In particular, changes in $PO_4^{3-}$ concentrations explained > 50 % of the variance in relative abundance for Bacteroidetes (fig. 9)."

**R#3:** Line 311: "8 major phyla" Are these 8 major phyla/classes listed somewhere in the paper? If not, please do so here.

**C:** The 8 major phyla/classes are Actinobacteria, Bacteroidetes, Alphaproteobacteria, Betaproteobacteria, Gammaproteobacteria, Cyanobacteria, Planctomycetes, and Verrucomicrobia.

**A:** For clarification we have now added text in Material and Methods section (Lines 241-245) to describe what is meant by the 8 major phyla/classes compared to the group "Others" which encompasses all other groups. Lines 241-245 reads as follows: "For all analyses on community composition we examined the following major eight phyla/classes: Actinobacteria, Bacteroidetes, Alphaproteobacteria, Betaproteobacteria, Gammaproteobacteria, Cyanobacteria, Planctomycetes, and Verrucomicrobia. All other phyla/classes were grouped together and defined as "Others"."

**R#3:** Line 320: "strong correlation" Can you please say in parentheses what constitutes a "strong correlation"?

**A:** The text now reads (Lines 386-391): "Although relative abundances of Gammaproteobacteria showed overall weak correlations with metabolic rates and environmental factors, the relative abundance of specific OTUs in this taxon, such as OTU 001410 and two Halioglobus OTUs (OTU 001149 and OTU 000045), displayed strong correlations (Pearson's r >0.5) with temperature, bacterial production and community respiration."

**R#3:** Line 321: Why "e.g."? Can you list all of the strong correlations, or only these few because there are too many?

**C:** See comment above. We have changed the text and removed "e.g."

**R#3:** Line 325: What is a substantial correlation? Please give a range, or an average, especially since the data is in the supplement. Listing something here allows the reader to better understand the relationship.

**A:** We have now added information in parenthesis on the level of correlation (Lines 393-396): "Betaproteobacteria affiliated with BAL58 showed in some cases a substantial correlation (Pearson's r >0.5) with DOC (OTU 001633, OTU 001481, OTU 000008 and OTU 001907) (fig. S3)."

**R#3:** Discussion: Line 356: The type of modeling exercise described in this section is valuable, and can be used to support a hypothesis, but I would caution against using the term "validate" in this case. It implies a level of certainty that I do not think can be reached in the current study.

**A:** We have changed the word "validate" by "support" as suggested by the reviewer.

**R#3:** Line 359-360: Could you please provide the coefficients for each parameter in this model, so that the reader can get an idea of the rate of change in BGE associated with each variable? They could be listed as a rate in parentheses after each parameter, for example. Were all of the parameters "significant" in the model? How was the model selected?

**C:** We have done all mixed effects models again to remove variables that had collinearity. Now we used DOC, DON, nitrate and phosphate concentration as variables and incubation day nested to season as random factor to account for temporal pseudo-replication.

**A:** We have done so. The text now reads (line 432-434): "Estimated BGE increased with nitrate (p < 0.003) and DOC concentration (p < 0.0009) and decreased with phosphate content (p < 0.02, mixed effects model, $R^2 = 0.79$)."

**R#3:** Line 368: Bacterial carbon demand was not measured in this study, rather the authors assume it based on community respiration. This statement should be amended to reflect the level of certainty that

can be supported by the data.

A: We have rephrased the sentence (lines 442-444): "(…) raised bacterial production at the same time that it reduced primary production, leading to more carbon being used by the microbial loop".

R#3: Line 369: The reduction in primary production does not lead to more carbon being used by the microbial loop. More carbon is used by the microbial loop because bacterial production (or respiration, which was not measured) increases.

C: The sentence tried to say that more carbon is used by the microbial loop because bacterial production increased at the same time that NCP decreased.

A: We re-wrote the sentence to make it more clear (see above).

R#3: Line 371: Could you provide a min-max range of the ratio of BP:NCP from your experiments to support this point (that the ecosystem moves towards heterotrophy)?

C: This statement is supported by the fact that NCP deceases at the same time than BP increases with DOC content from WWTP effluent. We calculated the ratio BP:NCP for our experiments and found that BP:NCP ratio tended to be higher in the treatments with WWTP effluent amendments (mean 1.56) than in treatments without WWTP addition (mean 0.66), but the differences between treatment types were not significant ($p > 0.05$). This points to a higher BP and lower NCP, moving the ecosystem towards heterotrophy with WWTP effluent inputs.

A: We include the results of the BP:NCP ratio here as suggested by the reviewer. The text now reads (lines 445-448): "This is supported by a higher BP:NCP ratio in treatments with addition of WWTP effluent (mean = 1.55 ± 045), compared to treatments without amendment (mean = 0.66 ± 0.31), although this differences are not significant ($p > 0.05$)."

R#3: Line 372: Increasing carbon flow into the microbial loop should not result in reduction in the transfer of carbon to higher trophic levels. Organic matter entering the microbial loop through bacterial uptake should still be returned to higher trophic levels through coupling with the traditional food chain. The paper by Wohlers refers to a decrease in carbon fixed by primary production being transferred to higher trophic levels (not organic C uptake by heterotrophs), and (as far as I can tell), the Berglund paper simply suggests that increased runoff (and thereby nutrient inputs) and temperature should favor a heterotrophic bacteria based food web and decrease production. Either way, I can't see why the authors conclude that increasing carbon flow into the microbial loop alone should result in a reduction in C transfer to higher trophic levels.

C: Berglund et al. said: "The food web efficiency, defined as mesozooplankton productivity per basal productivity (phytoplankton + bacteria), was 22% in the phytoplankton-based food web and 2% in the bacteria-based food web. We propose that climate change, with increased precipitation and river runoff in the Baltic Sea, might favor a bacteria-based food web and thereby reduce pelagic productivity at higher trophic levels." In bacterial-based food webs, due to smaller sizes of the resources and predators, these generally have more trophic levels than phytoplankton-based food webs. About 70% of the ingested carbon is lost at each trophic level due to respiration and sloppy feeding (Straile 1997), so, larger carbon losses are expected in bacteria-based food webs.

A: We have included a sentence to explain why increasing carbon flow into de microbial loop could result in a reduction of C transfer to higher trophic levels. The text now reads (Lines 451-456): "Bacteria-based food webs generally have lower food web efficiency due to the smaller sizes of the resources and predators, leading to more trophic levels than phytoplankton-based food webs. As around 70% of ingested carbon is lost at each trophic level due to respiration and sloppy feeding (Straile 1997), larger carbon losses are expected in bacteria-based food webs (Berglund et al., 2007)."

R#3: Line 379-380: It's not clear to me how this is related to the current study or discussion.

A: We have deleted this sentence as suggested by the reviewer.

R#3: Line 381-382: "A change in the planktonic community towards more heterotrophic communities" So far, this discussion has pointed out that rates of BP increased and the NCP decreased, with the addition of DOM. However, I don't think that there is evidence here that the community composition is shifting towards heterotrophy? Or, if there is evidence of this, it should be mentioned in the discussion here before lines 381-382.

A: We have re-written the sentence. The text now reads (lines 461-462): "Effluent inputs decreased GPP and NCP, resulting in a reduction of photosynthetic rates, declining oxygen production in the photic layer."

**R#3:** Line 382: While it is true that a reduction in photosynthetic rates would decrease oxygen production, I do not see clear evidence from this study that a shift towards heterotrophic communities is occurring, or that any reduction in photosynthetic rates would be the result of such a shift. In short, Line 381-382 make some assumptions that should be revisited and substantiated with data, if it exists. If it does not, then this discussion point should be reworded so that it is supported by the data.

    A: We have re-written the sentence. See above.

**R#3:** Line 390: "reducing the ecosystem capacity of removing nitrogen" Doesn't anoxia favor the removal of nitrogen (i.e. denitrification)?

    C: As the sediments become more reducing, more N is remobilized as ammonium and less as nitrate. The rates of denitrification slow down with the reduction of substrate (nitrate) and denitrification can be shut down at high respiration rates (Conley et al., 2009).  Several studies have demonstrated that in estuarine systems, denitrification displays a threshold-like behavior (Webster and Harris, 2004; Eyre and Ferguson, 2009), increasing to a maximum of carbon decomposition, and then decreasing as sediments become more reducing. Thus, when coastal and estuarine systems become hypoxic, there is a large risk that the loss of nitrogen will decrease (Smith & Hollibaugh, 1999), thus increasing the availability of DIN and acting as a positive feedback that increases the potential for eutrophication.

    A: We have now added a sentence to clarify how hypoxia can reduce the ecosystem capacity to remove nitrogen. The text now reads: "(…) as a consequence of the reduction of the substrate needed for denitrification (nitrate) when sediments become more reducing".

**R#3:** Line 390 - 393: The final two sentences here (lines 390-393) do not flow from the previous discussion about anoxia and eutrophication. The text in this section should be revised to make clear points and conclusions, which are supported by the data.

    A: We have moved the sentences to the end of the discussion section where we discuss effects of temperature.

**R#3:** Line 399: "disturbances" - Do you mean effluent inputs? what is meant by "disturbances"?

    **A:** We have re-written the sentence to make it more clear: "Our results showed that effluent inputs caused simultaneous shifts in community composition coupled with changes in metabolic rates."

**R#3:** Line 401: Can temperature be de-convoluted from season or other parameters? The changes in temperature weren't really "experimental" changes, but matched the in situ conditions at the time of sample collection, correct? I guess I don't really understand what is meant by "changes in temperature". And I still have the concerns listed above regarding the range of temperatures and influence of outlying temperature.

    **C:** The reviewer is correct that we cannot de- convolute temperature from season. However, despite the seasonal shifts in bacterial community composition, the effect of effluent inputs in summer affecting the community composition is noticeable. Please, see comment above on MANTEL tests and temperature.

**R#3:** Line 409 - 412: The sentence beginning "It is noteworthy" is not clear. The authors are suggesting that what changes in what relationships? Changes in the relationship between composition and function? What is the relationship between composition and function? I'm not sure where this sentence is going.

    **A:** We have done our best to clarify this sentence and added a reference so that it now reads as follows (Line 534-537): "In agreement, shifts in community composition can be closely linked with changes in community functioning, i.e. metabolic rates, (e.g. Bell et al. 2005 and Allison and Martiny 2008)."

Bell, T., et al. (2005). "The contribution of species richness and composition to bacterial services." Nature 436(7054): 1157-1160.

Allison, S. D. and J. B. Martiny (2008). "Colloquium paper: resistance, resilience, and redundancy in microbial communities." Proc Natl Acad Sci U S A 105 Suppl 1: 11512-11519.

**R#3:** Line 410: the link between community composition and function should be substantiated with a reference.

    **A:** See previous comment. Added references Bell et al. 2005, Allison and Martiny 2008.

**R#3:** Line 411-412: This is redundant with the discussion of theoretical BGE above.
  **A:** We have now deleted redundant text. See comment above.

**R#3:** Line 412-413: Lower diversity doesn't necessarily equate loss of function - aren't many functions redundant within a microbial community?
  **C:** We agree with the reviewer that lower diversity does not have to result in loss of function.
  **A:** We have therefore added a sentence with references to address this. The text reads (Lines 542-545): "Alternatively, lower richness and Shannon diversity index does not necessarily implicate loss of community functioning as previously observed in e.g. lake systems (Comte and del Giorgio 2011; Langenheder et al. 2005)"
Comte, J. and P. A. Del Giorgio (2011). "Composition influences the pathway but not the outcome of the metabolic response of bacterioplankton to resource shifts." PLoS One 6(9): e25266.
Langenheder, S., et al. (2005). "Weak coupling between community composition and functioning of aquatic bacteria." Limnology and Oceanography 50(3): 957-967.

**R#3:** Line 422: "caused responses" What responses? It would be more correct to say that these certain populations responded to effluent inputs. Also, didn't the Verrucomicrobia increase in the control? So how are changes in Verrucomicrobia associated with effluent inputs?
  **A:** We have now changed the text to the following (Lines 474-483): "In particular, verrucomicrobial and cyanobacterial populations responded in relative abundance to effluent inputs in summer. Thus, OTUs affiliated with Verrucomicrobia decreased in relative abundance in the treatments with effluent additions compared to controls. In contrast, the relative abundance of a few specific cyanobacterial populations increased upon enrichment (but less so in controls, i.e. the cyanobacterial growth was not only an effect of higher temperatures in the summer experiment). Generally, it is likely that the proliferation of cyanobacteria in the summer experiment is linked to the actual abundance of cyanobacteria, which is typically higher in summer, so that the "seeding" population for this taxon was higher."

**R#3:** Line 424: Nutrient inputs, or effluent inputs? The terminology used here is confusing, and I can't tell exactly what the authors are trying to conclude.
  **A:** The words "nutrient additions" have now been exchanged with "effluent inputs" throughout the text.

**R#3:** Line 430-431: "warming could increase cyanobacterial blooms" How did cyanos in this study respond to temperature?
  **C:** Relative abundances of Cyanobacteria were positively correlated with temperature.

**R#3:** Line 434-438: As written, this closing sentence (which should be used to drive home a major point of the current study) seems to focus on the results of a previous study instead. How does the current study and its findings support or add to the findings from the previous study? Also, the finding that "warming and effluent inputs increased planktonic respiration and bacterial production faster than primary production" is attributed to the previous study - I thought that this was a new conclusion of the current study? If not this, then what IS the new conclusion of the current study?
  **C:** We agree with the reviewer that we should use the closing sentence to give a take home message.
  **A:** We end the discussion section with the main conclusion of the present study. The text now reads (lines 573-577): "Here, we found that WWTP effluent inputs increased bacterial production at the same time that decreased net and gross primary production and community respiration. An parallel increase in bacterial production and decrease in primary production leads to more carbon being used by the microbial loop and may have consequences on the food web transfer efficiency."

**R#3:** Line 443: The conclusion that this leads to an increase in BGE is stated with more certainty than can be derived from the current study. It assumes that the decrease in CR is also a decrease in BR, but that may not be the case. The conjecture is ok, but should not be stated as fact.
  **A:** We have changed the sentence. The text now reads (line 582): "(…) which could lead to an increase in BGE".

**R#3:** Line 448-449: If cyanos increased in summer, how is this be linked to effluent inputs and not temperature? Also, I assume that "abundance" is "relative abundance"?

**C:** The response of Cyanobacteria was pronounced in the nutrient amendments (compared to controls) in the July experiment, while cyanobacterial responses were smaller at other times of the year. We think this is linked to the actual abundance of cyanobacteria, which is generally higher in summer, so that the "seeding" population is then much higher. Please see also additions made for Lines 474-483 to answer the reviewer's comment above.

**A:** We have now modified the text here in conclusions, Line 588-590 to emphasize that the Cyanobacteria increased in relative abundance for effluent amendments compared to controls: "In summer, the relative abundance of Cyanobacteria increased after effluent inputs (but less so in the controls)." We have changed "abundance" to "relative abundance", according to the reviewer.

**R#3:** Line 454: If cyanos are increasing due to effluent input, it is not clear to me how the conclusion that planktonic communities are shifting toward heterotrophic communities is made? Were the relative abundances of photo and heterotrophic organisms compared? Or is this based on rates of activity of the two groups? If the latter, this should be rephrased so that it does not lead the reader to conclude that the community structure is changing, and is responsible for a shift towards heterotrophy.

**C:** The conclusion that communities change towards heterotrophic communities with WWTP effluent inputs is made based on metabolic rates, with BP increasing and PP decreasing with effluent amendments.

**A:** We have rephrased the sentence to make it clear that we based this conclusion on the metabolic rates and not in bacterial community changes (lines 594-597): "Reductions of the OM content in wastewater treatment plant effluents are needed to reduce its potential negative consequences. Effluent inputs resulted in a reduction of photosynthetic rates, moving the system towards heterotrophy, decreasing oxygen production in the photic layer in the Baltic Sea."

**R#3:** Line 460: Low BR (not CR) compared to BP leads to high BGE. Since BR was not measured in this study, the authors should be careful regarding the level of certainty they assign to these conclusions. While interesting, any conclusion related to BGE is theoretical and should be used to guide further research, not stated as fact.

**A:** We have removed the sentence, as it was repetitive with previous text.

Tables and Figures:

**R#3:** Table 1: As the table contains more information than only nutrient content, a more descriptive caption should be used. Perhaps "physicochemical parameters" would be more appropriate. The caption should also reflect the number of replicates used to arrive at the listed standard errors. Chemical symbols for nutrients should be listed with proper superscripts, subscripts, and charges (throughout the text as well). If all other chemical species are listed in molar concentrations, DOC should be too. It is best to keep these consistent.

**A:** We have made all changes suggested by the reviewer.

**R#3:** Table 2: I notice here that the amount of P added is unknown for half of the treatments, which makes me question results related to changes in P. Can the authors address this please? Why is the carbon labeled as "TOC" if the samples were filtered as described in the Methods? How was C:N ratio calculated? Is it a ratio of DOC:DON? or DOC:DIN? Is it by mass, or moles?

**C:** P was below detection limit (30 ppb) in WWTP effluent in spring and summer. It was not unknown for half of the treatments, the phosphate concentration for each day and treatment is reported in supplementary table S1.

**A:** We have added a sentence to explain that for spring and summer experiments P was below detection limit (lines 256-257): "In summer and spring phosphate content in the effluent was below detection limit (30 µg/L, Table 2)." We have changed "TOC" by "DOC" and added an explanation on how C:N ratio was calculated at the table caption. The text now reads (lines 843-844): "C:N ratio is calculated as the ratio DOC:DON (moles)."

**R#3:** Table 3: Some explanation of all factors tested and the model selection parameters should appear in the Methods section. How was the best model chosen? Were all factors tested initially (and what are all the potential factors)? The random factor for "experiment" is referred to as "season" in the Methods section, is it not? Please change one or the other so it is consistent.

**C:** We agree with the reviewer that we should include an explanation on factors tested and model selection parameters in the methods section.

**A:** We have changed "experiment" by "season" in the table caption. We have also included information on factors tested and model selection in the methods section. The text now reads (lines

224-231): "Metabolic rates data from the four experiments were combined to test the relationship between the given metabolic rates and physicochemical parameters (Table 1) by mixed effects models. Physicochemical parameters were selected avoiding collinearity. Selected variables were DOC, DON, nitrate and phosphate concentration. We used DOC as a proxy for dissolved organic matter (DOM). Parameters were selected according to its significance. Variables were removed following its p value (i.e. variables with higher p value were removed first) until all parameters were significant. To account for pseudo-replication we used incubation day nested to season (i.e. experiment) as a random factor. The pseudo-$R^2$ of the models was calculated following Xu (2003)."

**R#3:** Table 4: I notice a lack of correlation with organics - does this not imply that the shifts in composition are not related to effluent? "specific environmental variables" Which environmental variables were used? It seems that there should exist a table, similar to tables 1 and 2, that gives the environmental parameters for each incubation (Tables 1 and 2 show environmental parameters at the collection site and in the effluent, respectively, correct?).

      **C:** The specific environmental variables and metabolic activities used in the MANTEL tests and listed in Table 4 in each row are the concentrations and values measured during the incubations. The correlation tests were thus performed across all samples, i.e. in all experiments and treatments. All environmental parameters for each incubation are detailed in Supplementary Information table S1.

      **A:** We have now modified Table 4 to also include MANTEL tests performed for each experiment individually to follow the format of Table 1 and 2. We have also removed NOx from the analyses and modified the table legend at Line 854-857; "Table 4. Results of MANTEL tests (Pearson's r) to examine if absolute shifts in bacterioplankton community composition were correlated to absolute changes specific environmental variables and metabolic rates measured in the incubations during the experiments. Significance is indicated in parenthesis."

      **C:** The lack of correlation with organics are likely due to differences between communities in different experiments, yet, we note in our new Table 4 that for example absolute changes in DOC is significantly correlated with absolute shifts in community composition; Pearson's r = 0.269 p=0.039 in July. Similarly, TDN was showing a tendency toward statistically significant differences with shifts in community composition also in July (Pearson's r=0.147 p=0.057) with a relatively high Pearson's r.

**R#3:** Figures 2, 4, and 5 should be of higher resolution. They appear blurry in the pdf.
      **A:** We have included those figures with higher resolution.

**R#3:** Figure 3: What is a whole model plot? What is the model? Please clarify this caption. And if whole model refers to some sort of model selection, it should be described.
      **A:** We have removed the word "whole" from the caption. We have re-phased the figure caption to better explain what it represents. The text now reads (lines 866-869): "Figure 3. Comparison of actual values and values predicted by the mixed effects model for (a) gross primary production (GPP), (b) community respiration (CR), (c) net community production (NCP) and (d) bacterial diversity. Black solid line represents the 1:1 line."

**R#3:** Figure 7: "A" is not labeled. It doesn't seem to be possible to see all of the treatments in figure A. I think only showing Figure B would be more informative.
      **A:** According to the suggestion by the reviewer we have removed panel A and kept the boxplots in panel B.

**R#3:** Figure 8: It would be easier to look at if we could see the controls first in the group for each time point and if the time points were separated somehow, perhaps by a small line
      **A:** We have now modified the figure according to the suggestions made by the reviewer.

**R#3:** Figure 9: What were the highest and lowest correlations? Cutting it off at 0.4 seems like it would bin together a lot of data, unless there are no strong correlations. If there are no correlations >0.4, this should be made apparent by this figure and/or its legend. Cutting it off at 0.4 doesn't tell me very much about what is going on here. I don't think that "NOx" was used previously in the paper, so shoud be defined in the figure legend. Nutrients are incorrectly labeled again (missing charges etc). I would suggest just writing out the names if it is difficult to properly add superscripts etc. in the software used for the figures.
      **C:** There is no threshold described in the figure referring to a maximum of 0.4. Actually the highest and lowest Pearson's r correlation are around 0.7 and -0.7, respectively. In this figure there are no positive or negative correlations that exceed 0.7.

**A:** We have modified the figure according to the suggestion made by the reviewer to use only names of variables, both here and in the supplementary figure S3. In Figure S3 we abbreviate the variables. We have removed NOx from the figures.

**R#3:** Minor comments:
Line 151: change "was" to "were" Line 199: change "sunlight" to "solar" Line 323: "MWH-UniP1" Add "related" to the end of this OTU designation. Line 329: Use the correct designation for phosphate Line 349: Add (BR) after bacterial respiration to define the acronym. Line 375: Citations needed for "some studies". Line 442: change "caused" to "was related to"

**A:** We have done all changes suggested by the reviewer.